# Universal probabilistic programming offers a powerful approach to statistical phylogenetics

Fredrik Ronquist [1,6 ✉], Jan Kudlicka [2,6], Viktor Senderov [1,6], Johannes Borgström[2], Nicolas Lartillot[3], Daniel Lundén [4], Lawrence Murray[5], Thomas B. Schön [2] & David Broman[4]

Statistical phylogenetic analysis currently relies on complex, dedicated software packages, making it difficult for evolutionary biologists to explore new models and inference strategies. Recent years have seen more generic solutions based on probabilistic graphical models, but this formalism can only partly express phylogenetic problems. Here, we show that universal probabilistic programming languages (PPLs) solve the expressivity problem, while still supporting automated generation of efficient inference algorithms. To prove the latter point, we develop automated generation of sequential Monte Carlo (SMC) algorithms for PPL descriptions of arbitrary biological diversification (birth-death) models. SMC is a new inference strategy for these problems, supporting both parameter inference and efficient estimation of Bayes factors that are used in model testing. We take advantage of this in automatically generating SMC algorithms for several recent diversification models that have been difficult or impossible to tackle previously. Finally, applying these algorithms to 40 bird phylogenies, we show that models with slowing diversification, constant turnover and many small shifts generally explain the data best. Our work opens up several related problem domains to PPL approaches, and shows that few hurdles remain before these techniques can be effectively applied to the full range of phylogenetic models.

[1] Department of Bioinformatics and Genetics, Swedish Museum of Natural History, Stockholm, Sweden. [2] Department of Information Technology, Uppsala University, Uppsala, Sweden. [3] Laboratoire de Biométrie et Biologie Evolutive, UMR CNRS 5558, Université Claude Bernard Lyon 1, Villeurbanne, France. [4] Department of Computer Science, KTH Royal Institute of Technology, Stockholm, Sweden. [5] Uber AI, San Francisco, CA, USA. [6] These authors contributed equally: Fredrik Ronquist, Jan Kudlicka, Viktor Senderov. ✉email: fredrik.ronquist@nrm.se

In statistical phylogenetics, we are interested in learning the parameters of models in which evolutionary trees—phylogenies —play an important part. Such analyses have a surprisingly wide range of applications across the life sciences[1–3]. In fact, the research front in many disciplines is partly defined today by our ability to learn the parameters of realistic phylogenetic models.

Statistical problems are often analyzed using generic modeling and inference tools. Not so in phylogenetics, where empiricists are largely dependent on dedicated software developed by small teams of computational biologists[3]. Even though these software packages have become increasingly flexible in recent years, empiricists are still limited to a large extent by predefined model spaces and inference strategies. Venturing outside these boundaries typically requires the help of skilled programmers and inference experts.

If it were possible to specify arbitrary phylogenetic models in an easy and intuitive way, and then automatically learn the latent variables (the unknown parameters) in them, the full creativity of the research community could be unleashed, significantly accelerating progress. There are two major hurdles standing in the way of such a vision. First, we must find a formalism (a language) that can express phylogenetic models in all their complexity, while still being easy to learn for empiricists (the modeling language expressivity problem). Second, we need to be able to generate computationally efficient inference algorithms from such model descriptions, drawing from the full range of techniques available today (the automated inference problem).

In recent years, there has been significant progress toward solving the expressivity problem by adopting the framework of probabilistic graphical models (PGMs)[4,5]. PGMs can express many components of phylogenetic models in a structured way, so that efficient Markov chain Monte Carlo (MCMC) samplers—the current workhorse of Bayesian statistical phylogenetics—can be automatically generated for them[5]. Other, more novel inference strategies are also readily applied to PGM descriptions of phylogenetic model components, as exemplified by recent work using STAN[6] or the new Blang framework[7].

Unfortunately, PGMs cannot express the core of phylogenetic models: the stochastic processes that generate the tree, and anything dependent on those processes. This is because the evolutionary tree has variable topology, while a PGM expresses a fixed topology. The problem even occurs on a fixed tree, if we need to express the possible existence of unobserved side branches that have gone extinct or have not been sampled. There could be any number of those for a given observed tree, each corresponding to a separate PGM instance.

Similar problems occur when describing evolutionary processes occurring on the branches of the tree. Many of the standard models considered today for trait evolution, such as continuous-time discrete-state Markov chains, are associated with an infinite number of possible change histories on a given branch. It is not always possible to represent this as a single distribution with an analytical likelihood that integrates out all change histories. Thus, it is sometimes necessary to describe the model as an unbounded stochastic loop or recursion over potential PGMs (individual change histories).

PGM-based systems may address these shortcomings by providing model components that hide underlying complexity. For instance, a tree may be represented as a single stochastic variable in a PGM-based model description[5]. An important disadvantage of this approach is that it removes information about complex model components from the model description. This forces users to choose among predefined alternatives instead of enabling them to describe how these model components are structured. Furthermore, computers can no longer "understand" these components from the model description, making it impossible to automatically apply generic inference algorithms to them. Instead, special-purpose code has to be developed manually for each of the components. Finally, hiding a complex model component, such as a phylogenetic tree, also makes dependent variables unavailable for automated inference. In phylogenetics, for instance, a single stochastic tree node makes it impossible to describe branch-wise relations between the processes that generated the tree and other model components, such as the rate of evolution, the evolution of organism traits, or the dispersal of lineages across space.

Here, we show that the expressivity problem can be solved using universal probabilistic programming languages (PPLs). A "universal PPL" is an extension of a Turing-complete general-purpose language, which can express models with an unbounded number of random variables. This means that random variables are not fixed statically in the model (as they are in a finite PGM) but can be created dynamically during execution.

PPLs have a long history in computer science[8], but until recently they have been largely of academic interest because of the difficulty of generating efficient inference machinery from model descriptions using such expressive languages. This is now changing rapidly thanks to improved methods of automated inference for PPLs[9–14], and the increased interest in more flexible approaches to statistical modeling and analysis. Current PPL inference algorithms provide state-of-the-art performance for many models but they are still quite inefficient for others. Improving PPL algorithms so that they can compete with manually engineered solutions for more problem domains is currently a very active research area.

To demonstrate the potential of PPLs in statistical phylogenetics, we tackle a tough problem domain: models that accommodate variation across lineages in diversification rate. These include the recent cladogenetic diversification rate shift (ClaDS) (ClaDS0–ClaDS2)[15], lineage-specific birth–death-shift (LSBDS)[16], and Bayesian analysis of macroevolutionary mixtures (BAMM)[17] models, attracting considerable interest among evolutionary biologists despite the difficulties in developing good inference algorithms for some of them[18].

Using WebPPL—an easy-to-learn PPL[9]—and Birch—a language with a more computationally efficient inference machinery[14]—we develop techniques that allow us to automatically generate efficient sequential Monte Carlo (SMC) algorithms from short descriptions of these models (~100 lines of code each). Although we found it convenient to work with WebPPL and Birch for this paper, we emphasize that similar work could have been done using other universal PPLs. Adopting the PPL approach allows us generate the first efficient SMC algorithms for these models, and the first asymptotically exact inference machinery for the full BAMM model. Among other benefits, SMC inference allows us to directly estimate the marginal likelihoods of the models, so that we can assess their performance in explaining empirical data using rigorous Bayesian model comparison. Taking advantage of this, we show that models with slowing diversification, constant turnover and many small shifts (all combined in ClaDS2) generally explain the data from 40 bird phylogenies better than alternative models. We end the paper by discussing a few problems, all seemingly tractable, which remain to be solved before PPLs can be used to address the full range of phylogenetic models. Solving them would facilitate the adoption of a wide range of novel inference strategies that have seen little or no use in phylogenetics before.

## Results

**Probabilistic programming.** Consider one of the simplest of all diversification models, constant rate birth–death (CRBD), in which lineages arise at a rate $\lambda$ and die out at a rate $\mu$, giving rise to a phylogenetic tree $\tau$. Assume that we want to infer the values of $\lambda$ and $\mu$ given some phylogenetic tree $\tau_{obs}$ of extant (now living) species that we have observed (or inferred from other data). In a Bayesian analysis, we would associate $\lambda$ and $\mu$ with prior distributions, and then

learn their joint posterior probability distribution given the observed value of $\tau$.

Let us examine a PGM description of this model, say in RevBayes[5] (Algorithm 1). The first statement in the description associates an observed tree with the variable `myTree`. The priors on `lambda` and `mu` are then specified, and it is stated that the tree variable `tau` is drawn from a birth–death process with parameters `lambda` and `mu` and generating a tree with leaves matching the taxa in `myTree`. Finally, `tau` is associated with (clamped to) the observed value `myTree`.

## Algorithm 1
**PGM description of the CRBD model**

```
1 myTree = readTrees("treefile.nex")
2
3 lambda ~ dnGamma(1, 1)
4 mu ~ dnGamma(1, 1)
5
6 tau ~ dnBirthDeath(lambda, mu, myTree.taxa)
7 tau.clamp(myTree)
```

There is a one-to-one correspondence between these statements and elements in the PGM graph describing the conditional dependencies between the random variables in the model (Fig. 1). Given that the conditional densities `dnGamma` and `dnBirthDeath` are known analytically, along with good samplers, it is now straightforward to automatically generate standard inference algorithms, such as MCMC, for this problem.

Unfortunately, a PGM cannot describe from first principles (elementary probability distributions) how the birth–death process produces a tree of extant species. The PGM has a fixed graph structure, while the probability of a surviving tree is an integral over many outcomes with varying topology. Specifically, the computation of `dnBirthDeath` requires integration over all possible ways in which the process could have generated side branches that eventually go extinct, each of these with a unique configuration of speciation and extinction events (Fig. 2). The integral must be computed by special-purpose code based on analytical or numerical solutions specific to the model. For the CRBD model, the integral is known analytically, but as soon as we start experimenting with more sophisticated diversification scenarios, as evolutionary biologists would want to do, computing the integral is likely to require dedicated numerical solvers, if it can be computed at all.

Universal PPLs solve the expressivity problem by providing additional expressive power over PGMs. A PPL model description is essentially a simulation program (or generative model). Each time the program runs, it generates a different outcome. If it is executed an infinite number of times, we obtain a probability distribution over outcomes. Thus, a PPL provides "a programmatic model description"[14].

A universal PPL provides two special constructs, one for drawing a random variable from a probability distribution, and one for conditioning a random variable on observed data. These special constructs are used by the PPL inference algorithms to manipulate executions of the program during inference. Many PPLs are embedded in existing programming languages, with these two special constructs added.

To use this approach, we need to write a PPL program so that the distribution over outcomes corresponds to the posterior probability distribution of interest. This is straightforward if we understand how to simulate from the model, and how to insert the constraints given by the observed data.

Assume, for instance, that we are interested in computing the probability of survival and extinction under CRBD for specific values of $\lambda$ and $\mu$, given that the process started at some time $t$ in the past. We will pretend that we do not know the analytical

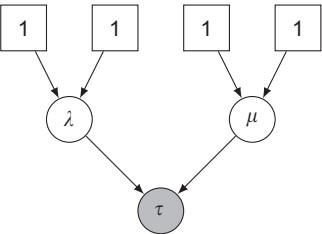

**Fig. 1 A probabilistic graphical model describing constant rate birth–death (CRBD).** The square boxes are fixed nodes (parameters of the gamma distributions) and the circles are random variables. The shaded variable ($\tau$) is observed, and $\lambda$, $\mu$ are latent variables to be inferred.

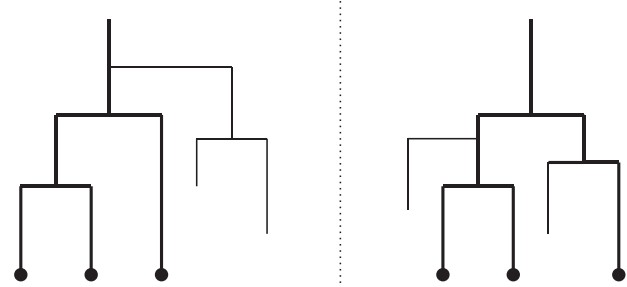

**Fig. 2 Phylogenetic trees generated by a birth–death process.** Two trees with extinct side branches (thin lines), each corresponding to the same observed phylogeny of extant species (thick lines). The trees illustrate just two examples of an infinite number of possible PGM expansions of the $\tau$ node in Fig. 1.

solution to this problem; instead we will use a PPL to solve it. WebPPL[9] is an easy-to-learn PPL based on JavaScript, and we will use it here for illustrating PPL concepts. WebPPL can be run in a web browser at http://webppl.org or installed locally (Supplementary Note Section 2.1). In WebPPL, the two special constructs mentioned above are: (1) the `sample` statement, which specifies the prior distributions from which random variables are drawn; and (2) the `condition` statement, conditioning a random variable on an observation. WebPPL provides a couple of alternatives to the `condition` statement, namely, the `observe` and `factor` statements. These are explained in Supplementary Note Section 3.3.

In WebPPL, we define a function `goesExtinct`, which takes the values of `time`, `lambda`, and `mu` corresponding to variables $t$, $\lambda$, and $\mu$, respectively (Algorithm 2). It returns `true` if the process does not survive until the present (that is, goes extinct) and `false` otherwise (survives to the present).

## Algorithm 2
**Basic birth-death model simulation in WebPPL**

```
1 var goesExtinct = function(time, lambda, mu) {
2     var waitingTime = sample(
3         Exponential({a: lambda + mu})
4     )
5
6     if (waitingTime > time) { return false }
7
8     var isSpeciation = sample(
9         Bernoulli({p: lambda / (lambda + mu)})
10    )
11
12    if (isSpeciation == false) { return true }
13
14    return goesExtinct(time - waitingTime, lambda, mu)
15        && goesExtinct(time - waitingTime, lambda, mu)
16 }
```

The function starts at some `time > 0` in the past. The `waitingTime` until the next event is drawn from an exponential distribution with rate `lambda + mu` and compared with `time`. If `waitingTime > time`, the function returns `false` (the process survived). Otherwise, we flip a coin (the `Bernoulli` distribution) to determine whether the next event is a speciation or an extinction event. If it is a speciation, the process continues by calling the same function recursively for each of the daughter lineages with the updated time `time - waitingTime`. Otherwise the function returns `true` (the lineage went extinct).

If executed many times, the `goesExtinct` function defines a probability distribution on the outcome space {`true`, `false`} for specific values of $t$, $\lambda$, and $\mu$. To turn this into a Bayesian inference problem, let us associate $\lambda$ and $\mu$ with gamma priors, and then infer the posterior distribution of these parameters assuming that we have observed a group originating at time $t = 10$ and surviving to the present. To do this, we combine the prior specifications and the conditioning on survival to the present with the `goesExtinct` function into a program that defines the distribution of interest (Algorithm 3).

## Algorithm 3
**CRBD model description in WebPPL**

```
1  var model = function() {
2     var lambda = sample(
3        Gamma({shape: 1, scale: 1})
4     )
5     var mu = sample(
6        Gamma({shape: 1, scale: 1})
7     )
8     var t = 10
9
10    condition(goesExtinct(t, lambda, mu) == false)
11
12    return [lambda, mu]
13 }
```

The `goesExtinct` function described above (Algorithm 2) uses unbounded stochastic recursion: the tree that we simulate in the program can in principle grow to infinite size. This effectively proves that the PPL defining this model, if it is to be used to simulate extinct side branches, must be a universal PPL. This, in turn, implies that a language that solves the expressivity problem in phylogenetics can also describe any phylogenetic model from which we can simulate using an algorithm. Adopting this approach thus allows a clean separation of model specification from inference. Of course, automated inference procedures now face the problem of executing complex universal PPL models on hardware with physical constraints, such as limited memory size. However, these challenges can typically be addressed using generic approaches that apply to arbitrary model descriptions, relieving both empiricists and algorithm developers from such concerns over technical implementation details.

Because of the popularity of PPLs in recent years, the term "probabilistic programming" is now often used to refer to the entire range of platforms, from universal PPLs to simple PGM frameworks. Unless we explicitly say otherwise, however, we will henceforth reserve "probabilistic programming" and "PPL" for platforms that implement universal modeling languages.

Inference in PPLs is typically supported by constructs that take a model description as input. Returning to the previous example, the joint posterior distribution is inferred by calling the built-in `Infer` function with the model, the desired inference algorithm, and the inference parameters as arguments (Algorithm 4).

## Algorithm 4
**Specifying inference strategy in WebPPL**

```
1  Infer({model: model, method: 'SMC', particles: 10000})
```

To develop this example into a probabilistic program equivalent to the RevBayes model discussed previously (Algorithm 1), we just need to describe the CRBD process along the observed tree, conditioning on all unobserved side branches going extinct (Supplementary Note Algorithms 2 and 3). The PPL specification of the CRBD inference problem is longer than the PGM specification because it does not use the analytical expression for the CRBD density. However, it exposes all the details of the diversification process, so it can be used as a template for exploring a wide variety of diversification models, while relying on the same inference machinery throughout. We will take advantage of this in the following.

**Diversification models**. The simplest model describing biological diversification is the Yule (pure birth) process[19,20], in which lineages speciate at rate $\lambda$ but never go extinct. For consistency, we will refer to it as constant rate birth (CRB). The CRBD model[21] discussed in the examples above adds extinction to the process, at a per-lineage rate of $\mu$.

An obvious extension of the CRBD model is to let the speciation and/or extinction rate vary over time instead of being constant[22], referred to as the generalized birth–death process. Here, we will consider variation in birth rate over time, keeping turnover ($\mu/\lambda$) constant, and we will refer to this as the time-dependent birth–death (TDBD) model, or the time-dependent birth (TDB) model when there is no extinction. Specifically, we will consider the function:

$$\lambda(t) = \lambda_0 e^{z(t_0 - t)}, \tag{1}$$

where $\lambda_0$ is the initial speciation rate at time $t_0$, $t$ is current time, and $z$ determines the nature of the dependency. When $z > 0$, the birth rate grows exponentially and the number of lineages explodes. The case $z < 0$ is more interesting biologically; it corresponds to a niche-filling scenario. This is the idea that an increasing number of lineages leads to competition for resources and—all other things being equal—to a decrease in speciation rate. Other potential causes for slowing speciation rates over time have also been considered[23].

The four basic diversification models—CRB, CRBD, TDB, and TDBD—are tightly linked (Fig. 3). When $z = 0$, TDBD collapses to CRBD, and TDB to CRB. Similarly, when $\mu = 0$, CRBD becomes equivalent to CRB, and TDBD to TDB.

In recent years, there has been a spate of work on models that allow diversification rates to vary across lineages. Such models can accommodate diversification processes that gradually change over time. They can also explain sudden shifts in speciation or extinction rates, perhaps due to the origin of new traits or other factors that are specific to a lineage.

One of the first models of this kind to be proposed was BAMM[17]. The model is a lineage-specific, episodic TDBD model. A group starts out evolving under some TDBD process, with extinction ($\mu$) rather than turnover ($\epsilon$) being constant over time. A stochastic process running along the tree then changes the parameters of the TDBD process at specific points in time. Specifically, $\lambda_0$, $\mu$, and $z$ are all redrawn from the priors at these switch points. In the original description, the switching process was defined in a statistically incoherent way; here, we assume that the switches occur according to a Poisson process with rate $\eta$, following a previous analysis of the model[18].

The BAMM model has been implemented in dedicated software using a combination of MCMC sampling and other

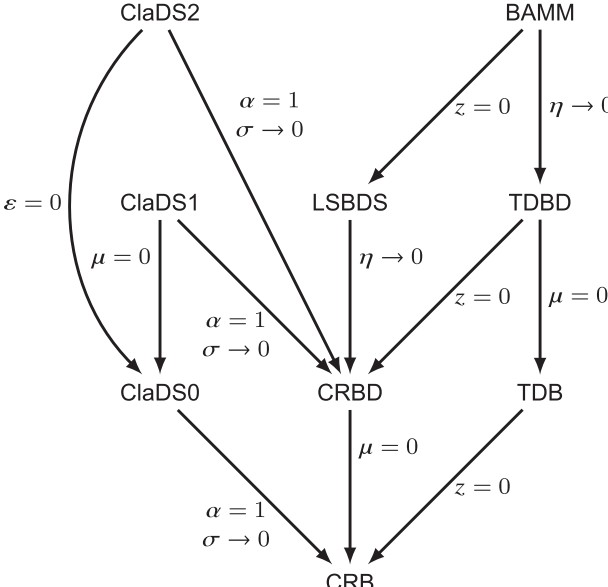

**Fig. 3 Relations between the diversification models considered in the paper.** Arrows and symbols mark the variable transformations needed to convert one diversification model into another.

numerical approximation methods[17,24]. The implementation has been criticized because it can result in severely biased inference[18]. To date, it has not been possible to provide asymptotically exact inference machinery for BAMM.

In a recent contribution, a simplified version of BAMM was introduced: the LSBDS model[16]. LSBDS is an episodic CRBD model, that is, it is equivalent to BAMM when $z = 0$. Inference machinery for the LSBDS model has been implemented in RevBayes[5] based on numerical integration over discretized prior distributions for $\lambda$ and $\mu$, combined with MCMC. The computational complexity of this solution depends strongly on the number of discrete categories used. If $k$ categories are used for both $\lambda$ and $\mu$, computational complexity is multiplied by a factor $k^2$. Therefore, it is tempting to simplify the model. We note that, in the empirical LSBDS examples given so far, $\mu$ is kept constant and only $\lambda$ is allowed to change at switch points[16]. When $z = 0$, BAMM collapses to LSBDS, and when $\eta \to 0$ it collapses to TDBD (Fig. 3). When $\eta \to 0$, LSBDS collapses to CRBD.

A different perspective is represented by the ClaDS models[15]. They map diversification rate changes to speciation events, assuming that diversification rates change in small steps over the entire tree. After speciation, each descendant lineage inherits its initial speciation rate $\lambda_i$ from the ending speciation rate $\lambda_a$ of its ancestor through a mechanism that includes both a deterministic long-term trend and a stochastic effect. Specifically:

$$\log \lambda_i \sim \mathcal{N}\left(\log\left(\alpha\lambda_a\right), \sigma^2\right). \qquad (2)$$

The $\alpha$ parameter determines the long-term trend, and its effects are similar to the $z$ parameter of TDBD and BAMM. When $\alpha < 1$, that is, $\log \alpha < 0$, the speciation rate of a lineage tends to decrease over time. The standard deviation $\sigma$ determines the noise component. The larger the value, the more stochastic fluctuation there will be in speciation rates.

The original ClaDS paper[15] focuses on the rate multiplier $m = \alpha \times \exp(\sigma^2/2)$ rather than on $\alpha$, but we prefer the $\alpha$ parameterization mainly because it allows us to specify a conjugate prior that makes SMC inference more efficient (Supplementary Note Section 3). As pointed out elsewhere[25], the dynamics of the ClaDS models is complex and differs considerably from superficially similar

models, such as the BAMM, TDBD, and TDB models (for further discussion of this point, see Supplementary Note Section 3).

There are three different versions of ClaDS, characterized by how they model $\mu$. In ClaDS0, there is no extinction, that is, $\mu = 0$. In ClaDS1, there is a constant extinction rate $\mu$ throughout the tree. Finally, in ClaDS2, it is the turnover rate $\epsilon = \mu/\lambda$ that is kept constant over the tree. All ClaDS models collapse to CRB or CRBD models when $\alpha = 1$ and $\sigma \to 0$ (Fig. 3). The ClaDS models were initially implemented in the R package RPANDA[26], using a combination of advanced numerical solvers and MCMC simulation[15]. A new implementation of ClaDS2 in Julia instead relies on data augmentation[25].

In contrast to previous work, where these models are implemented independently in complex software packages, we used PPL model descriptions (~100 lines of code each) to generate efficient and asymptotically correct inference machinery for "all" diversification models described above. The machinery we generate relies on SMC algorithms which, unlike classical MCMC, can also estimate the marginal likelihood (the normalization constant of Bayes theorem).

Estimating the marginal likelihood of a probability distribution that is only known up to a constant of proportionality is a hard problem in general. However, if we know how to sample from a similar distribution, classical importance sampling can provide a good estimate. The SMC algorithm is based on consecutive importance sampling from a series of probability distributions that change slowly toward the posterior distribution of interest. Thus, by piecing together the normalization constant estimates obtained in each of these steps, a good estimate of the marginal likelihood of the model is obtained essentially as a byproduct in the SMC algorithm[27,28]. Such series of similar probability distributions are not available naturally in the MCMC algorithm, but have to be constructed in more involved, computationally complex procedures, such as thermodynamic integration[29,30], annealed importance sampling[31], or stepping-stone sampling[32].

Using the SMC machinery, we then compared the performance of the different diversification models on empirical data by inferring the posterior distribution over the parameters of interest, and by conducting model comparison based on the marginal likelihood (Bayes factors). Specifically, we implemented the CRB, CRBD, TDB, TDBD, BAMM, LSBDS, and ClaDS0–ClaDS2 models in WebPPL and Birch. The model descriptions are provided at https://github.com/phyppl/probabilistic-programming. They are similar in structure to the CRBD program presented above.

**Inference strategies**. We used inference algorithms in the SMC family, an option available in both WebPPL and Birch. An SMC algorithm[33–35] runs many simulations (called particles) in parallel, and stops them when some new information, like the time of a speciation event or extinction of a side lineage, becomes available. At such points, the particles are subjected to "resampling", that is, sampling (with replacement) based on their likelihoods. SMC algorithms work particularly well when the model can be written such that the information derived from observed data can successively be brought to bear on the likelihood of a particle during the simulation. This is the case when simulating a diversification process along a tree of extant taxa, because we know that each "hidden" speciation event must eventually result in extinction of the unobserved side lineage. That is, we can condition the simulation on extinction of the side branches that arise (Supplementary Note Algorithm 3). Similarly, we can condition the simulation on the times of the speciation events leading to extant taxa.

Despite this, standard SMC (the bootstrap particle filter) remains relatively inefficient for these models, and is unlikely to

yield adequate samples of the posterior for real problems given realistic computational budgets. Therefore, we employed three new PPL inference techniques that we developed or extended as part of this study: alignment[36], delayed sampling[13], and the alive particle filter[37] (see "Methods").

**Empirical results.** To demonstrate the power of the approach, we applied PPLs to compare the performance of the nine diversification models discussed above for 40 bird clades (see "Methods" and Supplementary Note Table 6). The results (Supplementary Note Figs. 13–22) are well summarized by the four cases presented in Fig. 4. Focusing on marginal likelihoods (top row), we observe that the simplest models (CRB and CRBD), without any variation through time or between lineages, provide an adequate description of the diversification process for around 40% of the trees (Fig. 4, Alcedinidae). In the remaining clades, there is almost universal support for slowing diversification rates over time. Occasionally, this is not accompanied by strong evidence for lineage-specific effects (Fig. 4, Muscicapidae) but usually it is (Fig. 4, Accipitridae and Lari). In the latter case, the ClaDS models always show higher marginal likelihoods than BAMM and LSBDS, and this even for trees on which the latter do detect rate shifts (Fig. 4, Lari). Interestingly, ClaDS2 rarely outperforms ClaDS0, which assumes no extinction. More generally, models assuming no extinction often have a higher marginal likelihood than their counterparts allowing for it.

The parameter estimates (Fig. 4) show the conservative nature of the Bayes factor tests, driven by the relatively vague priors we chose on the additional parameters of the more complex models (Supplementary Note Fig. 2). However, even when complex models are marginally worse than simple or no-extinction models, there is evidence of the kind of variation they allow. For instance, the posterior distributions on $z$ and $\log \alpha$ suggest that negative time dependence is quite generally present. Similarly, more sophisticated models usually detect low levels of extinction when they are outperformed by extinction-free counterparts. For a more extensive discussion of these and other results, see Supplementary Note Section 10.

## Discussion

Universal PPLs provide stochastic recursion and dynamic creation of an unbounded number of random variables, which makes it possible to express virtually any interesting phylogenetic model. The expressiveness of PPLs is liberating for empiricists but it forces statisticians and computer scientists to approach the inference problem from a more abstract perspective. This can be challenging but also rewarding, as inference techniques for PPLs are so broadly applicable. Importantly, expressing phylogenetic models as PPLs opens up the possibility to apply a wide range of inference strategies developed for scientific problems with no direct relation to phylogenetics. Another benefit is that PPLs reduce the amount of manually written code for a particular inference problem, facilitating the task and minimizing the risk of inadvertently introducing errors, biases or inaccuracies. Our verification experiments (Supplementary Note Section 7) suggest that the light-weight PPL implementations of ClaDS1 and ClaDS2 provide more accurate computation of likelihoods than the thousands of lines of code developed in the initial implementation of these models[15].

Previous discussion on the relative merits of diversification models have centered around the results of simulations and arguments over biological realism[15–18,38], and it has been complicated by the lack of asymptotically correct inference machinery for BAMM[18,38]. Our most important contribution in this context is the refinement of PPL techniques so that it is now possible to implement correct and efficient parameter inference under a wide range of diversification models, and to compare their performance on real data using rigorous model testing procedures.

The PPL analyses of bird clades confirm previous claims that the ClaDS models provide a better description of lineage-specific diversification than BAMM[15]. Even when simpler models have higher likelihoods, the ClaDS models seem to pick up a consistent signal across clades of small, gradual changes in diversification rates. Like many previous studies[39], our analyses provide little or no support for extinction rates above 0. This might be due in part to systematic biases in the sampling of the leaves in the observed trees[40,41], a problem that could be addressed by extending our PPL model scripts (Supplementary Note Section 10.6). Such sampling biases can also give the impression of slowing diversification rates even when rates are constant, potentially explaining some of the support for negative values of $z$ and $\log \alpha$ in our posterior estimates. We want to emphasize, however, that there is a range of other possible explanations for these patterns[23]. The idea that lineage-specific variation in diversification rates might be responsible for low estimates of extinction rates in analyses using simpler models[42,43] finds little support in our results but we cannot exclude the possibility that even more sophisticated lineage-specific models than the ones considered here might provide evidence in favor of this hypothesis. An interesting observation is that models with constant turnover (as in ClaDS2) appear to fit empirical data better than those with constant extinction (as in ClaDS1), even though constant extinction has been commonly assumed in previous studies. A fascinating question that is now open to investigation is whether there remains evidence of occasional major shifts in diversification rates once the small gradual changes have been accounted for, something that could be addressed by a model that combines ClaDS- and BAMM-like features.

Our results show that PPLs can already now compete successfully with dedicated special-purpose software in several phylogenetic problem domains. Separately, we show how PPLs can be applied to models where diversification rates are dependent on observable traits of organisms (so-called state-dependent speciation and extinction models)[37]. Other problem domains that may benefit from the PPL approach already at this point include epidemiology[44], host-parasite co-evolution[45], and biogeography[46–49].

What is missing before it becomes possible to generate efficient inference machinery for the full range of phylogenetic models from PPL descriptions? Assume, for instance, that we would like to do joint inference of phylogeny (say from DNA sequence data) and diversification processes, instead of assuming that the extant tree is observed. This would seem to touch on the major obstacles that remain. We then need to extend our current PPL models so that they also describe the nucleotide substitution process along the tree, and condition the simulation on the observed sequences. To generate the standard MCMC machinery for sampling across trees from such descriptions, delayed sampling needs to be extended to summarize over ancestral sequences (Felsenstein's pruning algorithm)[50], and it should be applied statically through analysis of the script before the MCMC starts rather than dynamically. State-of-the-art MCMC algorithms for PPLs[12] must then be extended to generate computationally efficient tree samplers, such as stochastic nearest neighbor interchange[51]. Applying SMC algorithms for sampling across trees[52] is even simpler, it just requires delayed sampling to summarize over ancestral sequences. To facilitate the use of PPLs, we think it will also be important to provide a domain-specific PPL that is easy to use, while supporting both automatic state-of-the-art inference algorithms for phylogenetic problems as well as manual composition of novel inference strategies suited for this application domain. These all seem to be tractable problems, which we aim to address within the TreePPL project (treeppl.org).

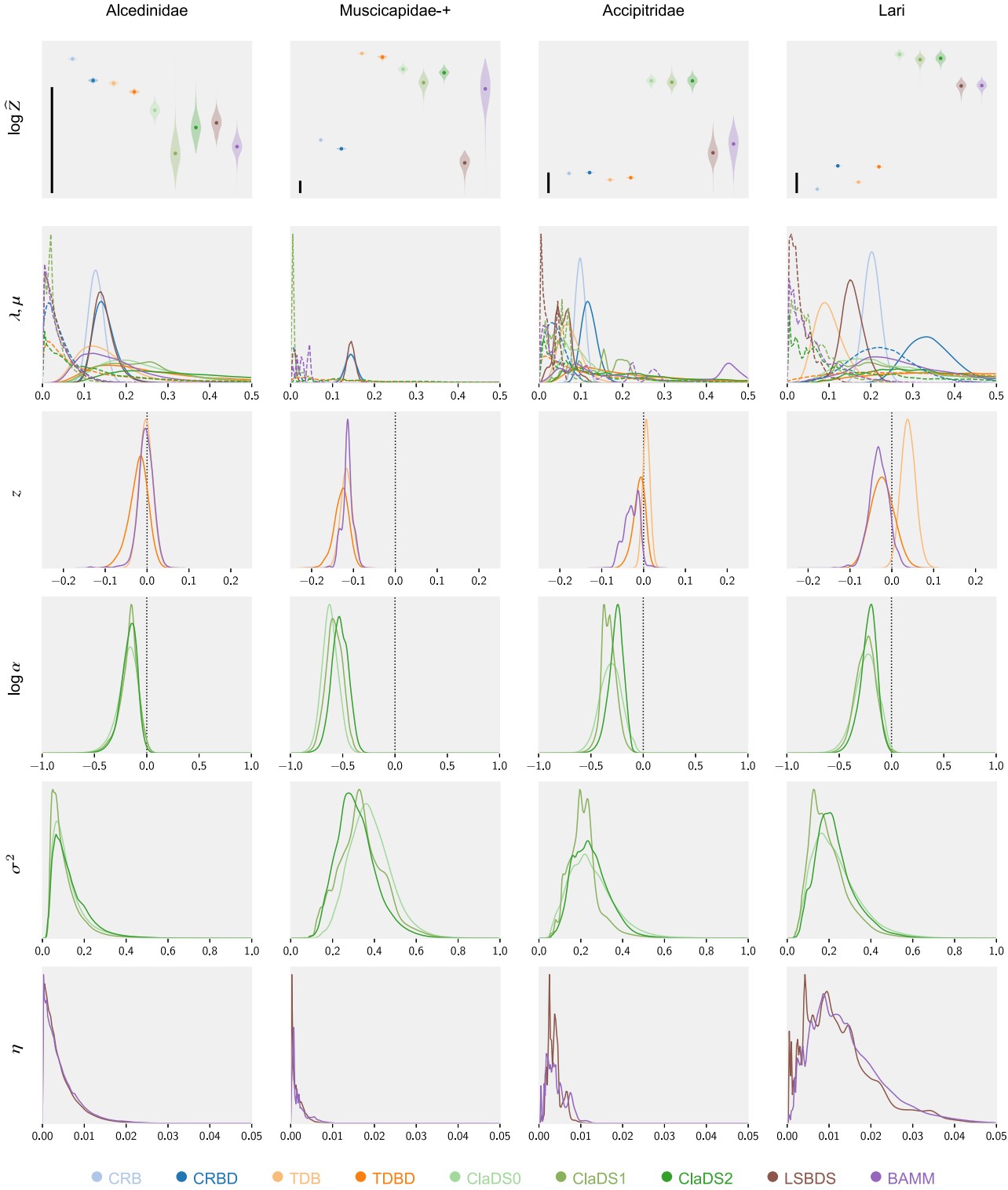

**Fig. 4 Comparison of diversification models for four bird clades exemplifying different patterns.** Alcedinidae: simple models are adequate; Muscicapidae: slowing diversification but no or weak lineage-specific effects; Accipitridae: gradual (ClaDS) lineage-specific changes in diversification; and Lari: evidence for both gradual (ClaDS) and for punctuated (BAMM and LSBDS) lineage-specific changes in diversification. The top row shows the estimated marginal likelihoods (log scale; violin plots with a dot marking the median estimate). A difference of 5 units (scale bar) is considered strong evidence in favor of the better model[67]. The remaining rows show estimated posterior distributions for different model parameters specified along the left margin. The $\mu$ distributions are shown with dashed lines, all other distributions with unbroken lines. The colors represent different models (see legend).

As the field of probabilistic programming is currently in a phase of intense experimentation, new PPL platforms—both universal and nonuniversal—are continuously presented and many existing ones are actively developed. Several of these platforms are likely to be useful for phylogenetic problems, not the least since they explore novel inference algorithms—such as automatic variational inference[53], adaptive Hamiltonian Monte Carlo[54], nonreversible parallel tempering[55] or sequential change

of measure[56]—that have only recently started to find their way into statistical phylogenetics[6,57,58]. Interesting platforms include not only RevBayes[5], specifically designed for phylogenetics, but also more general platforms such as STAN[59], Anglican[10], PyMC3[60], Edward[61], Pyro[62], and Blang[7]. We think that evolutionary biologists exploring these new tools will be excited by the expressivity of universal PPLs and the generality across model space of the automated inference solutions designed for them. With this in mind, we invite readers with an interest in computational methods to join us and others in developing languages and inference strategies supporting this powerful new approach to statistical phylogenetics.

## Methods

**PPL software and model scripts.** All PPL analyses described here used WebPPL version 0.9.15, Node version 12.13.1[9] and the most recent development version of Birch (as of June 12, 2020)[14]. We implemented all models (CRB, CRBD, TDB, TDBD, ClaDS0–ClaDS2, LSBDS, and BAMM) as explicit simulation scripts that follow the structure of the CRBD example discussed in the main text (Supplementary Note Section 5). We also implemented compact simulations for the four simplest models (CRB, CRBD, TDB, and TDBD) using the analytical equations for specific values of $\lambda$, $\mu$, and $z$ to compute the probability of the observed trees.

In the PPL model descriptions, we account for incomplete sampling of the tips in the phylogeny based on the $\rho$ sampling model[63]. That is, each tip is assumed to be sampled with a probability $\rho$, which is specified a priori. To simplify the presentation in this paper, we usually assume $\rho = 1$. However, the model scripts we developed support $\rho < 1$, and in the empirical analyses we set $\rho$ for each bird tree to the proportion of the known species included in the tree.

We standardized prior distributions across models to facilitate model comparisons (Supplementary Note Section 4, Supplementary Note Fig. 2). To simplify the scripts, we simulated outcomes on ordered but unlabeled trees, and reweighted the particles so that the generated density was correct for labeled and unordered trees (Supplementary Note Section 3.2). We also developed an efficient simulation procedure to correct for survivorship bias, that is, the fact that we can only observe trees that survive until the present (Supplementary Note Section 5.3).

**Inference strategies.** To make SMC algorithms more efficient on diversification model scripts, we applied three new PPL inference techniques: alignment, delayed sampling, and the alive particle filter. "Alignment"[36,64] refers to the synchronization of resampling points across simulations (particles) in the SMC algorithm. The SMC algorithms previously used for PPLs automatically resample particles when they reach `observe` or `condition` statements. Diversification simulation scripts will have different numbers and placements of hidden speciation events on the surviving tree (Fig. 2), each associated with a `condition` statement in a naive script. Therefore, when particles are compared at resampling points, some may have processed a much larger part of the observed tree than others. Intuitively, one would expect the algorithm to perform better if the resampling points were aligned, such that the particles have processed the same portion of the tree when they are compared. This is indeed the case; alignment is particularly important for efficient inference on large trees (Supplementary Note Fig. 3). Alignment at code branching points (corresponding to observed speciation events in the diversification model scripts) can be generated automatically through static analysis of model scripts[36]. Here, we manually aligned the scripts by replacing the statements that normally trigger resampling with code that accumulate probabilities when they did not occur at the desired locations in the simulation (Supplementary Note Section 6.1).

"Delayed sampling"[13] is a technique that uses conjugacy to avoid sampling parameter values. For instance, the gamma distribution we used for $\lambda$ and $\mu$ is a conjugate prior to the Poisson distribution, describing the number of births or deaths expected to occur in a given time period. This means that we can marginalize out the rate, and simulate the number of events directly from its marginal (gamma-Poisson) distribution, without having to first draw a specific value of $\lambda$ or $\mu$. In this way, a single particle can cover a portion of parameter space, rather than just single values of $\lambda$ and $\mu$. Delayed sampling is only available in Birch; we extended it to cover all conjugacy relations relevant for the diversification models examined here.

The "alive particle filter"[37] is a technique for improving SMC algorithms when some particles can "die" because their likelihood becomes 0. This happens when SMC is applied to diversification models because simulations that generate hidden side branches surviving to the present need to be discarded. The alive particle filter is a generic improvement on SMC, and it collapses to standard SMC with negligible overhead when no particles die. This improved version of SMC, partly inspired by our work on state-dependent speciation-extinction models[37], is only available in Birch.

**Verification.** To verify that the model scripts and the automatically generated inference algorithms are correct, we performed a series of tests focusing on the normalization constant (Supplementary Note Section 7). First, we checked that the

model scripts for simple models (CRB(D) and TDB(D)) generated normalization constant estimates that were consistent with analytically computed likelihoods for specific model parameter values (Supplementary Note Fig. 4). Second, we used the fact that all advanced diversification models (ClaDS0–2, LSBDS, and BAMM) collapse to the CRBD model under specific conditions, and verified that we obtained the correct likelihoods for a range of parameter values (Supplementary Note Fig. 6). Third, we verified for the advanced models that the independently implemented model scripts and the inference algorithms generated for them by WebPPL and Birch, respectively, estimated the same normalization constant for a range of model parameter values (Supplementary Note Fig. 7). Fourth, we checked that our normalization constant estimates were consistent with the RPANDA package[15,26] for ClaDS0–ClaDS2, and with RevBayes for LSBDS[5,16]. For these tests, we had to develop specialized PPL scripts emulating the likelihood computations of RPANDA and RevBayes. The normalization constant estimates matched for LSBDS (Supplementary Note Fig. 9) and for ClaDS0 (Supplementary Note Fig. 8); for ClaDS1 and ClaDS2, they matched for low values of $\lambda$ and $\mu$ (or $\epsilon$) but not for larger values (Supplementary Note Fig. 8). Our best-effort interpretation at this point is that the PPL estimates for ClaDS1 and ClaDS2 are more accurate than those obtained from RPANDA for these values (Supplementary Note Section 7.4). Finally, as there is no independent software that computes BAMM likelihoods correctly yet, we checked that our BAMM scripts gave the same normalization constant estimates as LSBDS under settings where the former model collapses to the latter (Supplementary Note Fig. 10).

**Data.** We applied our PPL scripts to 40 bird clades derived from a previous analysis of divergence times and relationships among all bird species[65]. The selected clades are those with more than 50 species (range 54–316) after outgroups had been excluded (Supplementary Note Table 6). We followed the previous ClaDS2 analysis of these clades[15] in converting the time scale of the source trees to absolute time units. The clade ages range from 12.5 to 66.6 Ma.

**Bayesian inference.** Based on JavaScript, WebPPL is comparatively slow, making it less useful for high-precision computation of normalization constants or estimation of posterior probability distributions using many particles. WebPPL is also less efficient than Birch because it does not yet support delayed sampling and the alive particle filter. Delayed sampling, in particular, substantially improves the quality of the posterior estimates obtained with a given number of particles. Therefore, we focused on Birch in computing normalization constants and posterior estimates for the bird clades.

For each tree, we ran the programs implementing the ClaDS, BAMM, and LSBDS models using SMC with delayed sampling and the alive particle filter as the inference method. We ran each program 500 times and collected the estimates of log Z from each run together with the information needed to estimate the posterior distributions. The quality of the normalization constant estimates (on the log scale) from these 500 runs was estimated using the standard deviation, as well as the relative effective sample size and the conditional acceptance rate (Supplementary Note Section 9). We initially set the number of SMC particles to 5000, which was sufficient to obtain high-quality estimates for all models except BAMM (Supplementary Note Table 7). We increased the number of particles to 20000 for BAMM to obtain estimates of acceptable quality for this model. For CRB, CRBD, TDB, and TDBD, we exploited the closed form for the likelihood in the programs. We used importance sampling with 10,000 particles as the inference method, and ran each program 500 times. This was sufficient to obtain estimates of very high accuracy for all models (Supplementary Note Table 7). The computational resources we used to obtain the results are specified in Supplementary Note Table 8.

**Visualization.** Visualizations were prepared with Matplotlib[66]. We used the collected data from all runs to draw violin plots for $\log \widehat{Z}$ as well as the posterior distributions for $\lambda$, $\mu$ (for all models), $z$ (for TDB, TDBD, and BAMM), $\log \alpha$ and $\sigma^2$ (for the ClaDS models), and $\eta$ (for LSBDS and BAMM). By virtue of delayed sampling, the posterior distributions for $\lambda$ and $\mu$ for all ClaDS models, as well as for BAMM and LSBDS, were calculated as mixtures of gamma distributions, the posterior distribution for $\log \alpha$ and $\sigma^2$ for all ClaDS models as mixtures of normal inverse gamma and inverse gamma distributions, and the posterior distribution for $\eta$ for BAMM and LSBDS as a mixture of gamma distributions. For the remaining model parameters, we used the kernel density estimation method. Exact plot settings and plot data are provided in the code repository accompanying the paper.

**Reporting summary.** Further information on research design is available in the Nature Research Reporting Summary linked to this article.

## Data availability

The dated phylogenetic trees used to compare the diversification models, together with full literature references, can be found at https://github.com/phyppl/probabilistic-programming, under the directory data. Supplementary information is available at https://github.com/phyppl/probabilistic-programming under the directory supplementary.

## Code availability

The WebPPL and Birch models can be found in the same repository, https://github.com/phyppl/probabilistic-programming, under the directories webppl and birch.

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

## Acknowledgements

We thank Lars Arvestad and Philippe Veber for their contributions to the early development of the ideas presented here, and Odile Maliet for extensive help with the RPANDA implementation of the ClaDS models. This work was supported by grants from the Swedish Research Council to J.B. and J.K. (No. 2013-4853) and to F.R. (No 2018-04620), by the Swedish Foundation for Strategic Research (SSF) via the project "ASSEMBLE" (Contract No.: RIT15-0012) to T.B.S., D.B., and J.K., and by the European Union's Horizon 2020 research and innovation program under the Marie Skłodowska-Curie grant agreement PhyPPL No. 898120 to V.S. The computations were performed on resources provided by the Swedish National Infrastructure for Computing (SNIC) at the National Supercomputer Centre (NSC).

## Author contributions

F.R. and N.L. initiated the project. All authors contributed to the further development of concepts and algorithms. F.R., J.K., and V.S. implemented algorithms, supported by D.L., J.B., L.M., N.L., T.B.S., and D.B. Verification experiments and empirical analyses were run by J.K. and V.S., who also generated most of the illustrations assisted by D.L., F.R., and J.B. The final paper was a joint effort.

## Competing interests

The authors declare no competing interests.
