## [Peer Review File · Communications Biology]

Reviewers' comments:

Reviewer #1 (Remarks to the Author):

Summary

The submitted manuscript outlines how to express phylogenetic processes within a probabilistic programming language and thereby show how to streamline phylogenetic modeling. The major benefits of adopting such a scheme relative to problem-bespoke samplers and packages are 1) added expressiveness, and 2) application of fairly generic inference methods to sample from posterior distributions over model parameters. An additional side benefit is the provenance of a corrected estimation technique to remedy a deficit in the inference code for the BMM model. Extensive numerical simulations were conducted with data from a range of sources, though all comparisons involved WebPPL and Birch, with no comparison to problem-specific platforms for model fitting. The main body of the article is augmented with an in-depth supplementary information that serves both as tutorial and review for a range of different probabilistic models.

Major comments

- I regard the authors' contribution to the body of research on statistical phylogenetic methodology to be worthy of publication and, for the most part, the numerical experiments performed make a strong argument for usage of a rich PPL for inference in phylogenetic models. The supplementary information, in particular, is a rich source of information and offers an excellent first tutorial for the use of PPLs. However, I suggest minor revisions to address shortcomings in the manuscript. My primary criticism is that this paper fails to adequately place the suitability of Birch and its related capacities in the larger sphere of PPLs and PGM frameworks.
- A key argument of the paper is that PPLs enable a generic, flexible way to express complicated processes and the main body of the article omits any reference to widely used tools for Bayesian modeling such as Stan (apparently) under the justification that they are not Turing complete PPLs, though some discussion is deferred to the supporting information. I see three problems with this omission: (1), a number of "PGMs" appear to be Turing complete despite a superficial lack of constructs deemed to be necessary prerequisites for membership in this class of programs. (2), I suspect that Turing completeness is not as important for model expressiveness as implied by lines 217 - 230. (3) This is a minor issue, but even if points (1) and (2) were to be incorrect, the paper would be more accurately characterized by the inclusion of the word "Universal" in the title so that potential readers with the requisite background would immediately be aware of the authors' assertions regarding the importance of Turing completeness. As it is right now, the article's title promises a fairly broad analysis relating probabilistic programming to statistical phylogenetics and this promise is not met by the text. I think that a revision of the title or an adjustment of the text is vitally needed.
- With regard to (1), my interpretation of the class descriptor "PGM" is that it includes modeling frameworks such as Stan and PyMC3 among many others. If not, I cannot think of a good reason not to at least mention their use in the main body of the text (as in Fourment and Darling, 2019) to make the paper more complete. I am very concerned that, in its current form, this article may give the reader the impression that "PGMs" are not useful because they cannot accommodate key phylogenetic stochastic processes. For example, PyMC3 and Stan are both capable of expressing stochastic branching and recursion; PyMC3 does it via the switch and scan programming constructs, respectively. Either explicit refutation of this point (if called for) or clarification of the class of PGMs relative to PPLs is recommended.
- In the event that Turing completeness is a critical feature of PPLs applied to statistical phylogenetics,

it would still be helpful to discuss comparisons with more feature-rich PPLs within the same class. I recommend at least a qualitative comparison in addition to citation in the SI with Edward and Pyro which are both Turing complete and which also have functionality for automatic differentiation, enabling automatic variational inference (Kucukelbir et al. 2016) and adaptive variants of Hamiltonian Monte Carlo (Hoffman et al. 2014). I think these could be extremely powerful tools for phylogenetic inference; even if subgraphs in a probabilistic model include discrete variables, such methods could be used as part of a larger message-passing scheme.

- Perhaps a more meaningful distinction from a computation point of view is whether or not all variables' memory requirements and size must be specified ahead of time. Note, however, that this is also a subtle and tricky point because many nonparametric Bayesian models (e.g. the Chinese restaurant process / hierarchical Dirichlet process) with potentially infinite numbers of parameters (and thus ostensibly benefitting most from representation within a more flexible, Turing-complete PPL) are still often implemented under the hood with a fixed memory footprint. Stan also enjoys a similar representational richness. The confusion on this subject is understandable because documentation on these software frameworks is not aimed at computer scientists but rather applied users for whom Turing completeness may not be an important concern. However, many commonly used probabilistic programming languages (including WebPPL) lean heavily upon Markov chain Monte Carlo methods such as Hamiltonian Monte Carlo leveraging gradients of the log posterior density. Since there appears to be some promise in using HMC for solving the big problems of statistical phylogenetics such as joint topological / parameter inference (Dinh et al. 2017), the completeness of this article would benefit from a more in-depth comparison of universal PPLs and what you refer to as PGM-based frameworks to help guide the reader. I have noticed that at least one of the models that is implemented in this paper has a discrete latent variable which precludes application of HMC in its current form without marginalization. If this is a generic feature of all models in the class of interest, then you may want to list this fact as a demerit of PGM frameworks such as Stan, though efforts to address this weakness are ongoing (Nishimura et al., 2020).

- With regard to the numerical experiments, I would like to see some summary statistic of the amount of computational effort required to reproduce your results. Rough estimates of the wall-clock time would be very useful.

Minor comments

- Line 43: Given my understanding of this work, it seems a bit odd to make this comment here. After all, do any of the proposed methods offer a solution to cross-tree inference? This appears to be at odds with line 468.
- Line 70: A more precise definition of "efficient" would be helpful here. WebPPL uses HMC and variational Bayes which may be more efficient in a loose sense even if SMC is currently one of few options available for dealing with the discrete combinatorial space of viable graphs.
- Line 351: There appears to be an extra space inside the parentheses.
- Line 368: If not included elsewhere, it would be helpful here to include a reference to the best review or tutorial for SMC methods that you can suggest.
- Line 387: A more precise definition of efficiency (i.e. in terms of raw computation or convergence of random variables as in the case of MCMC) would be appreciated here.
- Line 493: There appears to be prior research on using SMC to sample across trees. It is worth discussing their relevance. See Bouchard-Cote et al. (2012) and Wang et al. (2019) for more detail.
- Line 691: typo "edition edition" in this reference

References

Kucukelbir, A., Tran, D., Ranganath, R., Gelman, A., Blei, D.M., 2016. Automatic Differentiation

Variational Inference. arXiv:1603.00788 [cs, stat].

Hoffman, M.D., Gelman, A., 2014. The No-U-turn Sampler: Adaptively Setting Path Lengths in Hamiltonian Monte Carlo. *J. Mach. Learn. Res.* 15, 1593–1623.

Bouchard-Côté, A., Sankararaman, S., Jordan, M.I., 2012. Phylogenetic Inference via Sequential Monte Carlo. *Systematic Biology* 61, 579–593. <https://doi.org/10.1093/sysbio/syr131>

Dinh, V., Bilge, A., Zhang, C., Iv, F.A.M., n.d. Probabilistic Path Hamiltonian Monte Carlo 10.

Doucet, A., Godsill, S., Andrieu, C., n.d. On sequential Monte Carlo sampling methods for Bayesian filtering 12.

Fourment, M., Darling, A.E., 2019. Evaluating probabilistic programming and fast variational Bayesian inference in phylogenetics (preprint). *Bioinformatics*. <https://doi.org/10.1101/702944>

Wang, L., Wang, S., Bouchard-Côté, A., 2020. An Annealed Sequential Monte Carlo Method for Bayesian Phylogenetics. *Systematic Biology* 69, 155–183. <https://doi.org/10.1093/sysbio/syz028>

Nishimura, A., Dunson, D., Lu, J., 2017. Discontinuous Hamiltonian Monte Carlo for discrete parameters and discontinuous likelihoods. arXiv:1705.08510 [stat].

Reviewer #2 (Remarks to the Author):

The manuscript makes a compelling case that Probabilistic Programming Languages (PPLs) can be applied very fruitfully to certain problems in phylogenetic inference. Specifically, the authors examine the problem of analyzing differential rates of speciation using various models. The primary original advance that the authors make is the demonstration that a single unified programming approach may allow researchers to examine and compare multiple competing models using simpler and more general code than the current status quo in which separate implementations with very complex code bases are needed for each separate type of model. The manuscript is highly convincing that this new programming approach to computational phylogenetics has the potential to transform the field from one where biologists must learn and master a variety of specialized software programs for each separate type of problem to one where a single more general computing platform can be effective. The authors do an excellent job of validating their approach in the substantial supplemental work by showing that their results match those of alternative methods. I do think that this paper will influence others to explore the use of PPLs for other computational problems within phylogenetics. Furthermore, the sort of tree-based dependence structure that is at the heart of phylogenetic models is present in a variety of other problems, so there is potential that this work could influence computational problems in a wider area of disciplines.

I do raise a few caveats. The authors demonstrate the approach on a class of problems where the phylogeny (tree-shape of evolutionary relationships) is known and each example tree is rooted at a time that is assumed to be known without error, as each example is a different clade from a much larger bird phylogeny. There is no demonstration on how to approach a similar problem without the key information of the time of the most recent common ancestor, although, when aiming to estimate

relative speciation rates, this is probably not very important. A larger issue is the assumption of a known phylogeny. Most inference problems in phylogenetics do not assume a known tree, and in fact, determining this tree is frequently the primary question of interest. The authors express great optimism that the PPL approach demonstrated in this paper will be able to bridge the divide to an unknown tree in future work, but I am much more skeptical of this optimism. However, the present work does address a class of problems, identifying places in phylogenies that exhibit increases in the rate of speciation, which is an area of much current active research effort. The PPL approach may greatly lower the bar for developers to formulate and examine new modeling approaches. Another concern is the need that the authors had to adapt standard PPL methods to solve the problem they present. The changes they made demonstrate further innovation which is laudable, but does detract from the message that PPLs allow a simpler and more unified and general approach to addressing this class of problems. There are a few areas where the authors could offer more explanation. They note that the approaches using PPLs are comparatively slow, but they do not say by how much. This should be clarified. There are two important times to consider: how long does it take to do an analysis for a specific model and data set (where the existing methods are likely much faster) and how long does it take to develop and implement a new model, where I expect the PPL approach has substantial advantages. This latter comparison is necessarily not precise as it is measuring human effort, but some more detailed discussion on the tradeoff is warranted. Another item not mentioned are the required computational resources. How much memory and how many compute nodes are needed? I imagine that the requirements to maintain thousands of separate particles are substantial. Some detailed description of the computational requirements is warranted. One last comment is that the authors report the number of particles they use and indicate the need for a larger number of particles when using the BAMM model in one instance, but provide no guidance on how to select the number of particles and only limited guidance on how to determine if the choice is adequate.

Bret Larget

Reviewer #3 (Remarks to the Author):

In this paper, the authors demonstrate how Probabilistic Programming Languages (PPLs) can be used to develop an automatic inference machinery for fitting heterogeneous birth-death diversification models to reconstructed phylogenetic trees. They illustrate their approach by implementing fits of several recent diversification models, including ClaDS, LSBDS and BAMM. Importantly, this provides for the first time the possibility to compare these models directly using Bayes factors. They perform this model comparison on 40 bird clades and find that a model with many small rate variations (ClaDS) has higher marginal likelihood than a model with few large rate shifts (LSBDS, BAMM). The study is very well conducted, and accompanied with a thorough check of the implementation of the method. The approach has a great potential for the field. Whether the community will follow is a different question, but the approach is definitively novel and important. In addition, the study provides several new concrete possibilities in diversification modeling, in particular the ability to compare models that were previously not comparable. Only one of my comments requires re-running some analyses (in order to account for undersampling in the bird phylogenies). My other comments are remarks meant to improve the accessibility of the paper to the community. I overall very much enjoy this paper and I hope my comments will be useful.

Major Comments

1. The authors fixed $\rho=1$ (complete sampling) in their empirical analyses. Their justification (L530) is: 'Arguably, this is the relevant setting for the empirical analyses, as the selected trees comprise all or nearly all extant species.' However, in the Data section, the authors write that they used the same clades as in the Maliet et al. 2019 NEE paper. In that paper the Jetz trees that were used were those

with only the species for which there was molecular data, and these trees are not nearly complete. Given that the sampling fractions are known for these clades, I don't see a valid reason to assume complete sampling. This is important, as it could drive some of the diversification rate declines observed in the empirical analyses. Check also L1177 in the SI. This is not correct if the authors used the same trees as those used in Maliet et al. 2019, as the trees used there were those with only the species for which there was molecular data. These are not nearly complete.

2. If the authors want their PPL approach to take off in the community, it is important to make this paper as accessible as possible, to highlight the new possibilities and biological results made possible by the analyses, and to provide more information on the efficiency of the approach. I feel that this can be improved throughout.

- The authors could highlight the empirical analyses and associated results in the Abstract and Introduction. In the Abstract they write "we re-examine previous claims about the performance of the models.". This is probably going to be interpreted as "statistical performance" (power, type I error, bias in estimates) rather than as "ability of the models to accurately represent empirical data" (which is what I understand the authors want to say). I suggest to instead explicitly highlight the biological result that models with many small rate shifts have higher marginal likelihoods than models with few large shifts when fitted to bird phylogenies. This is a novel and important result and will appeal to the community. Similarly, in the last paragraph of the Introduction, the empirical application of the model comparison to bird clades should be explicitly mentioned.

- Another interesting empirical result that is not explicitly mentioned in the paper is that models with constant turnover (ClADS2) seem to always explain data better than models with constant extinction (ClADS1). This was expected from the Maliet et al. NEE 2019 paper, as summary statistics computed on trees simulated under ClADS2 matched those known for empirical trees better than under ClADS1, but not explicitly tested on empirical data in that paper. These two ways to model extinction correspond to two different conceptions of how extinction operates, and diversification models often assume constant extinction rather than constant turnover, so I think it is important to highlight this result in the text.

- A very nice new feature of the models' PPL implementation is that it can estimate the marginal likelihood, which allows model comparison. However, it is not clear which specific feature of the implementation makes this possible. The convergence of marginal likelihoods is usually very slow; the authors write 'This machinery relies on sophisticated Monte Carlo algorithms which, unlike classical MCMC, can also estimate the marginal likelihood (the normalization constant of Bayes theorem).' Is it just that SMC is more efficient than MCMC then? Maybe this is why the authors highlight that it is the first time that SMC (Sequential Monte Carlo) algorithms have been available for diversification models? As currently written, is not clear to me why the ability to implement SMC is so important, and which specific feature of the implementation renders the computation of marginal likelihoods tractable.

- Related to the later question is the question of diagnostic tests and computational efficiency. Little information is given about these two important aspects. A potential user (or model developer) will want to have an idea about this before choosing the PPL framework to analyze his/her data, or to develop a new model. Which convergence criteria were used in the paper and should be used in general? How many simulations/computational time are typically required? In the Methods, it is written that 5000 particles were used for all models, and 20,000 for BAMM, but what is the justification for these numbers?

- For someone not particularly familiar with the pros and cons of diverse programming languages (my case and I believe the case of many researchers in the field), it is hard to see what is so specific about PPL. L427-429 'Universal PPLs provide Turing-complete languages for model descriptions, which guarantees that virtually all interesting phylogenetic models can be expressed.' OK, but it seems that many languages could 'express' any phylogenetic model (take simply R to be provocative, or if it is not the case maybe what 'express' means need to be clarified?), and run a general SMC inference algorithm on these models. I agree that at the moment different diversification models are written in different software which makes their use and comparison difficult, but why could not a similar

homogenization effort be done in another language? I trust that there is something particular that I am missing, but I am afraid many readers might miss it as well, and anything the authors could do to help a reader get it would be appreciated.

- The authors could try to clarify the limits of probabilistic graphical models (PGMs) in a way that is more accessible to the community. This question is addressed in the Introduction paragraph starting from L40, but I did not understand what the authors meant before reading the PGM part of the Results (L119-135). As PGM is the model representation used in RevBayes, the authors could explain this in a more concrete way starting from the Introduction by specifying what RevBayes cannot do that PPLs could do? For example, "PGMs cannot express the stochastic processes that generate the tree" is opaque without more explanation, as it seems that RevBayes needs to express these processes in some way to estimate diversification rates from extant trees. Maybe specify something along the lines of "and therefore cannot fit diversification models to extant trees unless likelihoods can be computed"? Similarly, "it becomes impossible to describe relations between tree-generating processes and other model components, such as the rate of evolution, organism traits or biogeography." is opaque at first since state-dependent diversification models (for example) have been implemented in RevBayes.

3. The parameter that gives the temporal tendency in ClaDS is m ($m = \alpha \cdot \exp(\sigma^2/2)$), not α , given the expression for the mean of a lognormal distribution. This should be fixed throughout (main text L333-336, L420, Figure 4, SI Table 2 & Table 4, L337-340, 1079 etc.) so as to not introduce confusion about the interpretation of the ClaDS parameters.

4. Discussion about the lack of evidence for extinction (support for models without extinction and small extinction rate estimates). L463-464, the authors write 'This appears to be due in part to systematic biases in the sampling of the leaves in the observed trees', which is also the explanation given in Section S9.6. The authors focus on the potential role of diversified sampling, which indeed might be one of the possible explanations in general. However, it is unlikely to be the case here (rather complete phylogenies, diversified sampling potentially not an issue for birds), and other explanations have been given in the literature (there is a lot of literature on the subject). In particular, there has been discussions on the role of unaccounted for rate heterogeneities in biasing rate estimates. See for example Rabosky Evolution 2010 "Extinction rates should not be estimated from molecular phylogenies". In Morlon et al. PNAS 2011 "Reconciling molecular phylogenies with the fossil record", the authors show that substantial extinction can be recovered when rate heterogeneity is accounted for (in a model that is similar to BAMM but where rate shifts are fixed a priori based on taxonomy), while extinction rate estimates are close to zero when rate heterogeneity is not accounted for. This discussion seems particularly relevant in the context of the present paper, which implements rate heterogeneous models. Unfortunately, it seems that extinction rates are still low even when accounting for rate heterogeneity, but at least it seems that models that account for rate heterogeneity are associated with higher extinction rate estimates than models that do not? Discussion along these lines would be welcome.

Minor Comments

It seems that the paper by Barido-Sottani et al. 'A Multitype Birth-Death Model for Bayesian Inference of Lineage-Specific Birth and Death Rates' Syst Bio 2020 which presents a model very close (equivalent?) to the LSBDS model should somehow be cited in the paper.

L76 : « This is the first asymptotically exact inference machinery for BAMM." Does this statement still hold after the recent paper by Laudanno et al. "Detecting lineage-specific shifts in diversification: A proper likelihood approach" Syst Bio 2020? (I have to admit I haven't read that paper in detail yet). Check also L305-306, 448-450, 620-621, and L304-305 and 933, 1095 of the Supp Info concerning the analytical solution of BAMM.

L614-617: 'The normalization constant estimates matched for LSBDS (Supplementary Fig. 8) and for ClaDS0 (Supplementary Fig. 7) but not for ClaDS1 and ClaDS2.' This sentence seems a little bit too negative given results from Fig7. A sentence that would better represent results from Fig. 7 would be 'The normalization constant estimates matched for LSBDS (Supplementary Fig. 8) and for ClaDS0 (Supplementary Fig. 7); for ClaDS1 and ClaDS2, they matched for low values of lambda and mu (or epsilon), but not for larger values (Supplementary Fig. 7).'

Section S3.2. L217-220 'Our simulations are weighted with the appropriate conversion factor to generate the density for labelled and unoriented trees. Thus, the normalization constants we compute are directly comparable to the likelihoods computed using the standard analytical equations established for the simple diversification models, such as CRBD'. This comparison of normalization constant to the analytical likelihood is used in section S7 to verify the PPL scripts, and is certainly correct given the results, but could the authors explain this a little bit more to the reader? As I understand it with have: likelihood of labelled reconstructed tree = normalization constant * likelihood of unlabeled oriented tree. So how do we end up having normalization constant = likelihood of unlabeled oriented tree? This will also help understanding why the normalization constant provides model evidence (SI L495).

SI L353: "Upon examination of the empirical results published in the same paper, we concluded that this choice is overly conservative." Can the authors specify which empirical results led them to this conclusion? By overly conservative, do they mean that the choice of the prior in the Maliet et al. paper leads to an underestimation on rate heterogeneity?

SI L1043. Did the authors mean "positively related" instead of "inversely related"? (simple diversification models appear more adequate in old trees).

SI L1080: I guess the authors mean "higher marginal likelihoods"?

Here is a list of parts of the paper I did not really understand, in case the authors could clarify:

- L223: 'If such constructs are provided by PGM-based software, they are only executed when the model is initiated; they are not part of the model description itself.'
- SI L784-786: 'Since, for every execution of the program, there is a different number of hidden speciation events on each branch in the observed tree, this will cause the SMC particles to get out of sync at resampling points, so that we will be comparing particles that can be at very different points in the simulation.'
- SI Listing 3 is the description of a complete WebPPL script that one needs to understand to have a good idea of how the inference works. However, I found it hard to follow. I was particularly lost with the computation of the weights in the simulation functions. As mentioned on Lines 588-589, the simulation function does not return anything, but it weights the sampled parameters by conditioning the simulation on the observed tree, however I don't see how/where this weighting is done. Where does the computation of the weights $\ln\text{Prob1}$, $\ln\text{Prob2}$ and $\ln\text{Prob3}$ mentioned on Line 591-593 occur in the script?

Hélène Morlon

Summary of revision

We thank the reviewers for excellent and constructive comments, which were very helpful in improving the manuscript. In summary, the major changes are:

- We have positioned the paper more clearly in relation to other ongoing work on universal and non-universal PPLs. We also cover more of the PPL work related to phylogenetics or of particular interest to evolutionary biologists. Specifically:
 - The title now says "Universal probabilistic programming: ..." rather than just "Probabilistic programming:...", to clarify what the paper is about.
 - We have extended the text in the Introduction covering PGM-based approaches to statistical phylogenetics. We make it explicit that the cited studies represent examples of PGM-based approaches to phylogenetics, and we specify what platforms were used.
 - The last paragraph of the Discussion now provides many pointers to recent work on PPLs (universal and non-universal) and PPL-based inference algorithms of potential interest to phylogeneticists or evolutionary biologists.
 - The description of the expressivity of universal PPLs and the explanation of why this expressiveness is important in statistical phylogenetics have been improved and expanded (Introduction L43-L90; Results L188-L202, L281-L298). We now use more examples from statistical phylogenetics to make it easier for the community to follow the argumentation.
 - When we first describe universal PPLs in detail, we now emphasize that they are used for programmatic model descriptions and we refer to a mathematical definition of that concept (Results L194). We also clarify how universal PPLs differ from regular programming languages and why (L195-L202).
- We now summarize the most important results that emerged from our empirical analyses in the Abstract and in the last paragraph of the Introduction. We also discuss them in more detail in the Discussion and in the Supplementary material.
- We have added information about the computational resources we used to obtain the empirical results, and we provide diagnostics that measure the quality of the samples we obtained (Supplementary Section 9, and references to this section from the main text).
- Throughout the manuscript we have tried to improve the presentation to make the paper more accessible to the phylogenetics community, following suggestions from the reviewers.

Detailed responses to reviewer comments follow below.

Reviewer #1 (Remarks to the Author):

The submitted manuscript outlines how to express phylogenetic processes within a probabilistic programming language and thereby show how to streamline phylogenetic modeling. The major benefits of adopting such a scheme relative to problem-bespoke samplers and packages are 1) added expressiveness, and 2) application of fairly generic inference methods to sample from posterior distributions over model parameters. An additional side benefit is the provenance of a corrected estimation technique to remedy a deficit in the inference code for the BAMM model. Extensive numerical simulations were conducted with data from a range of sources, though all comparisons involved WebPPL and Birch, with no comparison to problem-specific platforms for model fitting. The main body of the article is augmented with an in-depth supplementary information that serves both as tutorial and review for a range of different probabilistic models.

We thank the reviewer for this positive summary of our work. However, there is a statement, which we do not understand. "Extensive numerical simulations were conducted with data from a range of sources, though all comparisons involved WebPPL and Birch, with no comparison to problem-specific platforms for model fitting". The relevant problem-specific platforms are RPANDA for the ClADS models and RevBayes for the LSBDS model. In the Supplementary information we do provide extensive comparisons between WebPPL and Birch on one hand, and these problem-specific platforms on the other. These comparisons are summarized in the Methods section of the main text. Please let us know if this comment refers to something else that we might have missed.

Major comments

I regard the authors' contribution to the body of research on statistical phylogenetic methodology to be worthy of publication and, for the most part, the numerical experiments performed make a strong argument for usage of a rich PPL for inference in phylogenetic models. The supplementary information, in particular, is a rich source of information and offers an excellent first tutorial for the use of PPLs. However, I suggest minor revisions to address shortcomings in the manuscript. My primary criticism is that this paper fails to adequately place the suitability of Birch and its related capacities in the larger sphere of PPLs and PGM frameworks.

A key argument of the paper is that PPLs enable a generic, flexible way to express complicated processes and the main body of the article omits any reference to widely used tools for Bayesian modeling such as Stan (apparently) under the justification that they are not Turing complete PPLs, though some discussion is deferred to the supporting information. I

see three problems with this omission: (1), a number of "PGMs" appear to be Turing complete despite a superficial lack of constructs deemed to be necessary prerequisites for membership in this class of programs. (2), I suspect that Turing completeness is not as important for model expressiveness as implied by lines 217 - 230. (3) This is a minor issue, but even if points (1) and (2) were to be incorrect, the paper would be more accurately characterized by the inclusion of the word "Universal" in the title so that potential readers with the requisite background would immediately be aware of the authors' assertions regarding the importance of Turing completeness. As it is right now, the article's title promises a fairly broad analysis relating probabilistic programming to statistical phylogenetics and this promise is not met by the text. I think that a revision of the title or an adjustment of the text is vitally needed.

We thank the reviewer for the appreciative comments and the constructive criticism. We agree that we could have described the relation between our work and other recent work on PPLs and PGM frameworks more clearly. We have now introduced 'Universal' in the title to clarify that the paper is about universal PPLs. Furthermore, we emphasize already when we first introduce universal PPLs that they extend Turing-complete programming languages, and we focus on the fact that this allows us to express models with an unbounded number of random variables, such as the diversification models discussed in the paper. We have also completely rewritten the passage on lines 217-230 in the old version of the manuscript (L281-L298 in revised manuscript). We now focus entirely on why a programmatic description of a diversification model requires a universal PPL. We also describe the universal PPL vision of separating model specification clearly from inference, and we point out that there are advantages of handling the physical limits of hardware in the inference step rather than in the modeling step.

We have removed the discussion of Turing completeness in PGM platforms. The discussion is difficult both because PGM model descriptions cannot necessarily be interpreted as programs to be executed, and because PGM-inspired platforms are evolving in expressiveness towards universal PPLs. We want to focus the paper on the features that are required to express all phylogenetic models and to apply automated inference to them, not on a potentially lengthy discussion of which platforms may or may not support these features at the moment.

Nevertheless, in our view there is a clear difference in expressiveness of universal PPLs and PGM-based systems like Stan and PyMC3. As far as we understand, Stan cannot express general stochastic recursion. The dimension of the model is set in the parameters block, and no other random variables may be generated in the model, so all loops in the model have deterministic termination behavior. The 'scan' construct in PyMC3 iterates a function over an existing piece of data, so it always terminates. Thus, it is weaker than general recursion, and

does not provide for a Turing-complete language. However, as mentioned above, we do think that this discussion is outside of the scope of the current paper.

With regard to (1), my interpretation of the class descriptor "PGM" is that it includes modeling frameworks such as Stan and PyMC3 among many others. If not, I cannot think of a good reason not to at least mention their use in the main body of the text (as in Fourment and Darling, 2019) to make the paper more complete. I am very concerned that, in its current form, this article may give the reader the impression that "PGMs" are not useful because they cannot accommodate key phylogenetic stochastic processes. For example, PyMC3 and Stan are both capable of expressing stochastic branching and recursion; PyMC3 does it via the switch and scan programming constructs, respectively. Either explicit refutation of this point (if called for) or clarification of the class of PGMs relative to UPPLs is recommended.

In the paper, we do emphasize the added expressivity and generality of universal PPLs compared to (classical) PGMs but it was not our intention to downplay the importance or usefulness of PGM frameworks in statistical phylogenetics. After all, a couple of us (FR and NL) have spent considerable time and effort on developing PGM approaches to statistical phylogenetics in the RevBayes project precisely because we think such approaches are important and useful.

We have reread the text with the concern expressed by the reviewer in mind. We can partly agree with the reviewer that our text can give the impression that PGMs are less useful in phylogenetics than they are. However, we did cite the paper mentioned by the reviewer (Fourment and Darling, 2019), which uses STAN and variational inference for (parts of) phylogenetic inference problems, when we discussed PGMs in the Introduction of the original text (L38-L39). We also cited the paper by Bouchard-Côté et al (2019), which presents a new PGM ('declarative modeling') framework (Blang) that explicitly refers to phylogenetic inference as part of the target domain. What was missing from the original version of the text, we think, was enough context for readers to understand the relevance of these citations. We now make it explicit that the references are examples of PGM-based approaches to phylogenetic problems, and we also mention the software platforms used in these papers.

We have also added a few sentences to the last paragraph of the Discussion, pointing readers to the potential use in phylogenetics of a range of PPLs, both universal and non-universal. Here, we explicitly cite RevBayes (the PGM platform for statistical phylogenetics co-authored by two of us) as well as Anglican, STAN, PyMC3, Blang, Anglican, Edward and Pyro. We also mention some of the promising new inference algorithms these frameworks support.

In the event that Turing completeness is a critical feature of PPLs applied to statistical phylogenetics, it would still be helpful to discuss comparisons with more feature-rich PPLs within the same class. I recommend at least a qualitative comparison in addition to citation in the SI with Edward and Pyro which are both Turing complete and which also have functionality for automatic differentiation, enabling automatic variational inference (Kucukelbir et al. 2016) and adaptive variants of Hamiltonian Monte Carlo (Hoffman et al. 2014). I think these could be extremely powerful tools for phylogenetic inference; even if subgraphs in a probabilistic model include discrete variables, such methods could be used as part of a larger message-passing scheme.

We agree with the reviewer that other PPLs, both universal and non-universal, can be very useful for phylogenetic problems. As mentioned above, we have now added a few sentences to the end of the Discussion, pointing out some of the most exciting recent developments in PPLs. We explicitly mention automatic variational inference and adaptive Hamiltonian Monte Carlo as new and promising directions that are explored within these PPL frameworks, in addition to sequential change of measure and non-reversible parallel tempering. When we first mention WebPPL and Birch in the main text, we also emphasize that we found it convenient to work with these universal PPLs for this paper, but that similar work could have been done using other universal PPLs.

We would like to refrain from a more comprehensive review of PPL platforms—universal or not—and their features. It would be quite challenging to attempt such a review for many reasons. Limitations that occur in one software version may not be there in the next version, so one would have to be meticulous with the criteria used to decide on whether features are present or absent in a software package to make the comparison fair. This would necessitate communication with all the software authors involved; ideally, they would all be invited as co-authors. Completing a review at this point would also be challenging because the field is developing so rapidly, almost guaranteeing that the review would be out of date before it was published.

The key points we try to convey in the paper are at a more conceptual level: universal PPLs can support efficient and automated inference for phylogenetic models that require Turing-complete programmatic descriptions with unbounded numbers of stochastic variables. This has been clarified in several places in the manuscript. For instance, when we describe universal PPLs, we now emphasize that they use programmatic model descriptions, and we provide a reference to a paper where this concept is defined mathematically (L194, see also above). We also give more examples of phylogenetic models that require unbounded numbers of stochastic variables (L43-L90).

We do not think it is critical to show in the paper which software frameworks that could, in principle, support this kind of approach now or in the near future. As far as we are aware, the inference techniques we use—alignment, delayed sampling and the alive particle filter—are not all available as automated features in any universal PPL at the moment. Birch probably comes closest in that it supports two of them, delayed sampling and the alive particle filter. We emphasize that we base our work on WebPPL and Birch because we find them convenient for our purposes, but we emphasize that similar work could have been done using other universal PPLs.

Perhaps a more meaningful distinction from a computation point of view is whether or not all variables' memory requirements and size must be specified ahead of time. Note, however, that this is also a subtle and tricky point because many nonparametric Bayesian models (e.g. the Chinese restaurant process / hierarchical Dirichlet process) with potentially infinite numbers of parameters (and thus ostensibly benefitting most from representation within a more flexible, Turing-complete PPL) are still often implemented under the hood with a fixed memory footprint. Stan also enjoys a similar representational richness. The confusion on this subject is understandable because documentation on these software frameworks is not aimed at computer scientists but rather applied users for whom Turing completeness may not be an important concern. However, many commonly used probabilistic programming languages (including WebPPL) lean heavily upon Markov chain Monte Carlo methods such as Hamiltonian Monte Carlo leveraging gradients of the log posterior density. Since there appears to be some promise in using HMC for solving the big problems of statistical phylogenetics such as joint topological / parameter inference (Dinh et al. 2017), the completeness of this article would benefit from a more in-depth comparison of universal PPLs and what you refer to as PGM-based frameworks to help guide the reader. I have noticed that at least one of the models that is implemented in this paper has a discrete latent variable which precludes application of HMC in its current form without marginalization. If this is a generic feature of all models in the class of interest, then you may want to list this fact as a demerit of PGM frameworks such as Stan, though efforts to address this weakness are ongoing (Nishimura et al., 2020).

As explained above, we think a detailed review of which PPL frameworks that are capable of handling particular model aspects, such as discrete latent variables, or that support particular inference algorithms, such as HMC, is outside of the scope of the current paper. Adaptive HMC definitely holds promise in tackling some of the most challenging phylogenetic problems, such as joint parameter and topology inference, but so does several other new inference algorithms that have been presented recently. Indeed, one of the advantages of

adopting a universal PPL approach, we argue, is that it will facilitate the adoption of such new algorithms in statistical phylogenetics, and we stressed this in the final paragraph of the Discussion in the original version of the manuscript. We now also cite the Dinh et al (2017) paper in this paragraph, among the new and promising inference algorithms mentioned there.

In our view, the challenge of running a Turing-complete program describing a probabilistic model on hardware with limited memory is something that is best handled in the automated inference step. This allows a cleaner separation of model specification from inference. It also allows generic strategies to be employed in addressing these physical constraints, rather than relying on specific implementations for particular distributions or model types. This is explained in the revised manuscript (L281-L298), as mentioned above.

With regard to the numerical experiments, I would like to see some summary statistic of the amount of computational effort required to reproduce your results. Rough estimates of the wall-clock time would be very useful.

This is an excellent point, raised by the other reviewers as well. We have now added a section in the supplementary material (Section 9, "Efficiency of inference algorithms") that specifies the computational resources we used to generate the empirical results.

Minor comments

Line 43: Given my understanding of this work, it seems a bit odd to make this comment here. After all, do any of the proposed methods offer a solution to cross-tree inference? This appears to be at odds with line 468.

Universal PPLs can readily express the phylogenetic inference problem, which should be obvious from our paper we hope. In principle, automated inference of phylogeny is available for these descriptions, it is just not efficient enough to be of any practical use at this point. To further demonstrate the relevance of being able to express variable topology in PPL approaches to phylogenetics, we have extended the Discussion of the cross-tree inference problem (L486 (not 468) and following in the original version of the manuscript) by specifying how little remains to be done before we can support efficient SMC inference of phylogeny in universal PPLs. Specifically, this boils down to automating the discrete-state version of delayed sampling for the standard character-state evolution models used in phylogenetics, a relatively straightforward problem to solve. This should remove any apparent conflict between the Discussion and the statement on L43 of the original version of the

manuscript, where we refer to the importance of being able to express the phylogeny inference problem in a PPL model description.

To convince ourselves and the reviewer that efficient inference of tree topology is not difficult to support in universal PPLs, we hardcoded the discrete-state version of delayed sampling for a phylogenetic substitution model into WebPPL. Using this approach, we succeeded in automatically generating efficient SMC inference of topology from universal probabilistic programs. We plan to present these results separately, when the entire inference procedure is automated and implemented in TreePPL. We cannot cite these unpublished results at the current time, and we think that further discussion of the tree inference problem in the universal PPL context is outside the scope of the current paper. Nevertheless, we hope that these recent experiments will help convince the reviewer that the comment on line 43 of the original manuscript version, where we refer to the importance of being able to express the phylogeny inference problem, is relevant, and that it is not at odds with the fact that some work remains to be done before efficient automated tree inference can be fully supported in universal PPLs.

Line 70: A more precise definition of "efficient" would be helpful here. WebPPL uses HMC and variational Bayes which may be more efficient in a loose sense even if SMC is currently one of few options available for dealing with the discrete combinatorial space of viable graphs.

We referred to computational efficiency. In the revised manuscript, we have clarified this point.

Line 351: There appears to be an extra space inside the parentheses.

There is a missing tilde sign here. This has been corrected now.

Line 368: If not included elsewhere, it would be helpful here to include a reference to the best review or tutorial for SMC methods that you can suggest.

This is an excellent suggestion. We have added a few references to papers that provide good introductions to SMC methods.

Line 387: A more precise definition of efficiency (i.e. in terms of raw computation or convergence of random variables as in the case of MCMC) would be appreciated here.

Here we refer to the quality of the sample of the posterior that can be obtained for real problems given reasonable computational resources. This is specified in the revised manuscript.

Line 493: There appears to be prior research on using SMC to sample across trees. It is worth discussing their relevance. See Bouchard-Cote et al. (2012) and Wang et al. (2019) for more detail.

This is correct. We have revised this text so that it covers automated generation of both SMC and MCMC inference algorithms for sampling across trees from universal PPL model descriptions. In the revised version of this paragraph, we also cite recent work on SMC and other recent inference algorithms for phylogeny inference.

Line 691: typo "edition edition" in this reference

Corrected.

Reviewer #2 (Remarks to the Author):

The manuscript makes a compelling case that Probabilistic Programming Languages (PPLs) can be applied very fruitfully to certain problems in phylogenetic inference. Specifically, the authors examine the problem of analyzing differential rates of speciation using various models. The primary original advance that the authors make is the demonstration that a single unified programming approach may allow researchers to examine and compare multiple competing models using simpler and more general code than the current status quo in which separate implementations with very complex code bases are needed for each separate type of model. The manuscript is highly convincing that this new programming approach to computational phylogenetics has the potential to transform the field from one where biologists must learn and master a variety of specialized software programs for each separate type of problem to one where a single more general computing platform can be effective. The authors do an excellent job of validating their approach in the substantial

supplemental work by showing that their results match those of alternative methods. I do think that this paper will influence others to explore the use of PPLs for other computational problems within phylogenetics. Furthermore, the sort of tree-based dependence structure that is at the heart of phylogenetic models is present in a variety of other problems, so there is potential that this work could influence computational problems in a wider area of disciplines.

We thank the reviewer for this positive summary of our work.

I do raise a few caveats. The authors demonstrate the approach on a class of problems where the phylogeny (tree-shape of evolutionary relationships) is known and each example tree is rooted at a time that is assumed to be known without error, as each example is a different clade from a much larger bird phylogeny. There is no demonstration on how to approach a similar problem without the key information of the time of the most recent common ancestor, although, when aiming to estimate relative speciation rates, this is probably not very important. A larger issue is the assumption of a known phylogeny. Most inference problems in phylogenetics do not assume a known tree, and in fact, determining this tree is frequently the primary question of interest. The authors express great optimism that the PPL approach demonstrated in this paper will be able to bridge the divide to an unknown tree in future work, but I am much more skeptical of this optimism.

As mentioned above, we have recently demonstrated that it is possible to do automated SMC inference of phylogeny from universal probabilistic programs, and we hope to present this work in a separate paper in the near future. Thus, we think there is reason for optimism on the larger issue.

Relaxing the assumption that the time of the most recent common ancestor is known is straightforward. However, it requires that one chooses a reasonable prior for the clock rate, which is not a trivial problem. It also requires conversion of the time trees of Jetz et al to trees with branch lengths specified in terms of the amount of evolutionary change. For these reasons, we think that adding this component to the empirical analyses would be outside of the scope of the current paper. As noted by the reviewer, accommodating uncertainty in the time of the most recent common ancestor is not likely to have a major effect on the results, as our focus is on the relative rates and how they change over time and across lineages in different diversification models. The main effect of relaxing the assumption of a fixed time for the most recent common ancestor is likely to be on the absolute rates of speciation and extinction, that is, the scaling of the lambda and mu values. Specifically, we expect that there

will be some added uncertainty about the absolute rates of these parameters. Other aspects of the analysis results should remain essentially the same.

However, the present work does address a class of problems, identifying places in phylogenies that exhibit increases in the rate of speciation, which is an area of much current active research effort. The PPL approach may greatly lower the bar for developers to formulate and examine new modeling approaches. Another concern is the need that the authors had to adapt standard PPL methods to solve the problem they present. The changes they made demonstrate further innovation which is laudable, but does detract from the message that PPLs allow a simpler and more unified and general approach to addressing this class of problems.

In the revised version of the text, we emphasize that universal PPLs solve the modeling language expressivity problem, but acknowledge that there is still a fair amount of work left before there are efficient PPL-based inference algorithms for all types of phylogenetic models. This is an active area for PPL researchers, with the space of supported models expanding all the time (L100-L103 in the revised manuscript).

There are a few areas where the authors could offer more explanation. They note that the approaches using PPLs are comparatively slow, but they do not say by how much. This should be clarified. There are two important times to consider: how long does it take to do an analysis for a specific model and data set (where the existing methods are likely much faster) and how long does it take to develop and implement a new model, where I expect the PPL approach has substantial advantages. This latter comparison is necessarily not precise as it is measuring human effort, but some more detailed discussion on the tradeoff is warranted. Another item not mentioned are the required computational resources. How much memory and how many compute nodes are needed? I imagine that the requirements to maintain thousands of separate particles are substantial. Some detailed description of the computational requirements is warranted.

We agree with the reviewer that this is an important aspect that we should have covered in the paper. We have now added a new section (Section 9, "Efficiency of inference algorithms") to the Supplementary material, which details the computational resources we used. Trying to address the second point raised by the reviewer (how long it takes to develop and implement a model) would be quite interesting but also very challenging, with no established methodology of which we are aware. In the Discussion, we do present the approximate number of lines of

code required by our model scripts, and compare it to the lines of code used in RPANDA for the ClaDS models (L539-544). Getting much beyond this would require a detailed analysis that would ideally involve experiments with human subjects. This would clearly be outside the scope of the current paper.

One last comment is that the authors report the number of particles they use and indicate the need for a larger number of particles when using the BMM model in one instance, but provide no guidance on how to select the number of particles and only limited guidance on how to determine if the choice is adequate.

Again, we agree with the reviewer that these topics should have been covered in the paper. The new section in the Supplementary material (Section 9, "Efficiency of inference algorithms") presents some relevant measures of the quality of the posterior sample obtained with SMC methods (or other Monte Carlo methods), and apply them to the empirical results we obtained for the bird data. In the same section, we also justify the increased number of particles used for the BMM analyses with reference to these quality measures.

Reviewer #3 (Remarks to the Author):

In this paper, the authors demonstrate how Probabilistic Programming Languages (PPLs) can be used to develop an automatic inference machinery for fitting heterogeneous birth-death diversification models to reconstructed phylogenetic trees. They illustrate their approach by implementing fits of several recent diversification models, including ClaDS, LSBDS and BMM. Importantly, this provides for the first time the possibility to compare these models directly using Bayes factors. They perform this model comparison on 40 bird clades and find that a model with many small rate variations (ClaDS) has higher marginal likelihood than a model with few large rate shifts (LSBDS, BMM). The study is very well conducted, and accompanied with a thorough check of the implementation of the method. The approach has a great potential for the field. Whether the community will follow is a different question, but the approach is definitively novel and important. In addition, the study provides several new concrete possibilities in diversification modeling, in particular the ability to compare models that were previously not comparable. Only one of my comments requires re-running some analyses (in order to account for undersampling in the bird phylogenies). My other comments are remarks meant to improve the accessibility of the paper to the community. I overall very much enjoy this paper and I hope my comments will be useful.

We thank the reviewer for these encouraging words, and appreciate the constructive and very useful comments on the manuscript.

Major Comments

1. The authors fixed $\rho=1$ (complete sampling) in their empirical analyses. Their justification (L530) is: ‘Arguably, this is the relevant setting for the empirical analyses, as the selected trees comprise all or nearly all extant species.’ However, in the Data section, the authors write that they used the same clades as in the Maliet et al. 2019 NEE paper. In that paper the Jetz trees that were used were those with only the species for which there was molecular data, and these trees are not nearly complete. Given that the sampling fractions are known for these clades, I don’t see a valid reason to assume complete sampling. This is important, as it could drive some of the diversification rate declines observed in the empirical analyses. Check also L1177 in the SI. This is not correct if the authors used the same trees as those used in Maliet et al. 2019, as the trees used there were those with only the species for which there was molecular data. These are not nearly complete.

We agree that the choice to fix ρ to 1 was unfortunate. We have recomputed all empirical results with the appropriate ρ values, and we have updated the manuscript accordingly. The statements on L530 in the main paper and on L1177 in the SI have been corrected.

2. If the authors want their PPL approach to take off in the community, it is important to make this paper as accessible as possible, to highlight the new possibilities and biological results made possible by the analyses, and to provide more information on the efficiency of the approach. I feel that this can be improved throughout.

We agree and appreciate the help in improving the presentation of our work.

- The authors could highlight the empirical analyses and associated results in the Abstract and Introduction. In the Abstract they write “we re-examine previous claims about the performance of the models.”. This is probably going to be interpreted as “statistical performance” (power, type I error, bias in estimates) rather than as “ability of the models to accurately represent empirical data” (which is what I understand the authors want to say). I suggest to instead explicitly highlight the biological result that models with many small rate shifts have higher marginal likelihoods than models with few large shifts when fitted to bird phylogenies. This is a novel and important result and will appeal to the community. Similarly,

in the last paragraph of the Introduction, the empirical application of the model comparison to bird clades should be explicitly mentioned.

These are very good points. We have now rewritten the second half of the Abstract to more effectively describe to the community what we accomplish in the paper, and we also highlight the empirical results. We have clarified that we assess how well the models explain data, not the statistical performance as defined by the reviewer. We have also rewritten the last paragraph of the Introduction along the lines suggested by the reviewer. We now explicitly mention the empirical application of the model comparison to bird clades, and we briefly summarize the results from these analyses. We have also slightly expanded the section of the Discussion in the main text that focuses on the empirical results.

- Another interesting empirical result that is not explicitly mentioned in the paper is that models with constant turnover (ClaDS2) seem to always explain data better than models with constant extinction (ClaDS1). This was expected from the Maliet et al. NEE 2019 paper, as summary statistics computed on trees simulated under ClaDS2 matched those known for empirical trees better than under ClaDS1, but not explicitly tested on empirical data in that paper. These two ways to model extinction correspond to two different conceptions of how extinction operates, and diversification models often assume constant extinction rather than constant turnover, so I think it is important to highlight this result in the text.

This is something that we also noted, but failed to point out in the manuscript. We now briefly mention this result in the Abstract and in the last paragraph of the Introduction. We also mention it now in the Discussion.

- A very nice new feature of the models' PPL implementation is that it can estimate the marginal likelihood, which allows model comparison. However, it is not clear which specific feature of the implementation makes this possible. The convergence of marginal likelihoods is usually very slow; the authors write 'This machinery relies on sophisticated Monte Carlo algorithms which, unlike classical MCMC, can also estimate the marginal likelihood (the normalization constant of Bayes theorem).' Is it just that SMC is more efficient than MCMC then? Maybe this is why the authors highlight that it is the first time that SMC (Sequential Monte Carlo) algorithms have been available for diversification models? As currently written, is not clear to me why the ability to implement SMC is so important, and which specific feature of the implementation renders the computation of marginal likelihoods tractable.

This is an important but subtle point that we are happy to explain further. In the revised version of the text, we try to describe the essential difference between SMC and MCMC algorithms with respect to the estimation of the marginal likelihood in an accessible and non-technical way. We also provide several references to key papers that discuss this topic in more detail.

- Related to the later question is the question of diagnostic tests and computational efficiency. Little information is given about these two important aspects. A potential user (or model developer) will want to have an idea about this before choosing the PPL framework to analyze his/her data, or to develop a new model. Which convergence criteria were used in the paper and should be used in general? How many simulations/computational time are typically required? In the Methods, it is written that 5000 particles were used for all models, and 20,000 for BAMM, but what is the justification for these numbers?

We agree that this is a serious omission, touched upon by the other reviewers as well. We have now added a new section (Section 9, "Efficiency of inference algorithms") to the Supplementary material, presenting relevant diagnostic tests and applying them to our empirical analyses. Based on quality assessment of the results, we also justify why we increased the number of particles for some analyses. Finally, in the same section, we also present data on the computational resources we used to obtain the empirical results.

- For someone not particularly familiar with the pros and cons of diverse programming languages (my case and I believe the case of many researchers in the field), it is hard to see what is so specific about PPL. L427-429 'Universal PPLs provide Turing-complete languages for model descriptions, which guarantees that virtually all interesting phylogenetic models can be expressed.' OK, but it seems that many languages could 'express' any phylogenetic model (take simply R to be provocative, or if it is not the case maybe what 'express' means need to be clarified?), and run a general SMC inference algorithm on these models. I agree that at the moment different diversification models are written in different software which makes their use and comparison difficult, but why could not a similar homogenization effort be done in another language? I trust that there is something particular that I am missing, but I am afraid many readers might miss it as well, and anything the authors could do to help a reader get it would be appreciated.

In the revised version of the manuscript, we specify how PPLs differ from regular programming languages when we first introduce universal PPLs. Specifically, we point out that PPLs include special constructs for sampling random variables from appropriate distributions and for conditioning random variables on observed data, and we explain that it is these special constructs that make it possible to apply generic inference algorithms to probabilistic programs. We also point out that many PPLs are based on regular programming languages, and simply extend them by adding these two types of special constructs.

- The authors could try to clarify the limits of probabilistic graphical models (PGMs) in a way that is more accessible to the community. This question is addressed in the Introduction paragraph starting from L40, but I did not understand what the authors meant before reading the PGM part of the Results (L119-135). As PGM is the model representation used in RevBayes, the authors could explain this in a more concrete way starting from the Introduction by specifying what RevBayes cannot do that PPLs could do? For example, “PGMs cannot express the stochastic processes that generate the tree” is opaque without more explanation, as it seems that RevBayes needs to express these processes in some way to estimate diversification rates from extant trees. Maybe specify something along the lines of “and therefore cannot fit diversification models to extant trees unless likelihoods can be computed”? Similarly, “it becomes impossible to describe relations between tree-generating processes and other model components, such as the rate of evolution, organism traits or biogeography.” is opaque at first since state-dependent diversification models (for example) have been implemented in RevBayes.

This is an important passage and we thank the reviewer for pointing out that it is difficult to understand, and for suggesting possible improvements. We have completely rewritten this passage to clarify the PGM shortcomings that we try to address in the paper using universal PPLs. We use examples from RevBayes and phylogenetics throughout the new version of this passage to make it easier for the community to follow the argumentation. We specifically address how RevBayes tackles complex phylogenetic model components, and why the universal PPL approach offers important advantages.

*3. The parameter that gives the temporal tendency in ClaDS is m ($m = \alpha * \exp(\sigma^2/2)$), not α , given the expression for the mean of a lognormal distribution. This should be fixed throughout (main text L333-336, L420, Figure 4, SI Table 2 & Table 4, L337-340, 1079 etc.) so as to not introduce confusion about the interpretation of the ClaDS parameters.*

We see several advantages of representing the temporal tendency in the ClaDS models using the parameter alpha instead of m. First, the noise parameter, sigma, is given on the logarithm scale, so it makes sense to also give the temporal tendency on the log scale (alpha) rather than on the linear scale (m). Second, alpha is more directly comparable to the z parameter in the TDB(D) models and in BAMM than m. Finally, alpha allows us to formulate a convenient conjugate prior that supports delayed sampling, making SMC inference more efficient. In the revised version of the paper, we spell out these reasons for preferring the alpha parameterization over the m parameterization in the Supplementary information, and we refer to this text from the main text when we introduce the ClaDS models.

4. Discussion about the lack of evidence for extinction (support for models without extinction and small extinction rate estimates). L463-464, the authors write ‘This appears to be due in part to systematic biases in the sampling of the leaves in the observed trees’, which is also the explanation given in Section S9.6. The authors focus on the potential role of diversified sampling, which indeed might be one of the possible explanations in general. However, it is unlikely to be the case here (rather complete phylogenies, diversified sampling potentially not an issue for birds), and other explanations have been given in the literature (there is a lot of literature on the subject). In particular, there has been discussions on the role of unaccounted for rate heterogeneities in biasing rate estimates. See for example Rabosky Evolution 2010 “Extinction rates should not be estimated from molecular phylogenies”. In Morlon et al. PNAS 2011 “Reconciling molecular phylogenies with the fossil record”, the authors show that substantial extinction can be recovered when rate heterogeneity is accounted for (in a model that is similar to BAMM but where rate shifts are fixed a priori based on taxonomy), while extinction rate estimates are close to zero when rate heterogeneity is not accounted for. This discussion seems particular relevant in the context of the present paper, which implements rate heterogeneous models. Unfortunately, it seems that extinction rates are still low even when accounting for rate heterogeneity, but at least it seems that models that account for rate heterogeneity are associated with higher extinction rate estimates than models that do not? Discussion along these lines would be welcome.

We agree with the reviewer that this is interesting previous work that would be valuable to comment on. Interestingly, our results do not appear to support the idea that heterogeneity in diversification rates is causing the unrealistically low extinction rate estimates in simple models. The extinction rate estimates in our analyses are quite similar for comparable models that do and that do not accommodate lineage-specific variation in diversification rates, sometimes there is even a tendency for the differences to run contrary to the hypothesis. For instance, extinction rates are usually estimated to be lower by LSBDS, accommodating

lineage-specific variation in diversification rates, than by CRBD, the corresponding model without lineage-specific variation. Similarly, ClaDS1 is usually associated with similar or lower extinction rate estimates than CRBD. ClaDS2 and BAMM are best compared to TDBD. The initial extinction rate estimates are similar in these models; they are not distinctly higher in ClaDS2 (results more uncertain for BAMM). Thus, our results suggest that lineage-specific variation in diversification rates is not an important explanation for unrealistically low extinction rate estimates. Of course, it is still possible that other types of rate heterogeneity than the ones accommodated in the examined models could play a role. We now comment on these results in the Discussion in the main text, including the disclaimer, and we describe these results in more detail in the Supplementary material.

Minor Comments

It seems that the paper by Barido-Sottani et al. ‘A Multitype Birth–Death Model for Bayesian Inference of Lineage-Specific Birth and Death Rates’ Syst Bio 2020 which presents a model very close (equivalent?) to the LSBDS model should somehow be cited in the paper.

The model introduced in this paper is related to but not equivalent to the LSBDS model. In short, it introduces a restriction on the number of diversification model categories. To speed up likelihood computations, it samples across histories of shifts among these model categories, and it also assumes that there are no shifts on extinct side branches. The paper is now discussed in the Supplementary material in connection with the presentation of the LSBDS model. To save some space, we do not discuss it in the main paper.

L76 : “This is the first asymptotically exact inference machinery for BAMM.” Does this statement still hold after the recent paper by Laudanno et al. “Detecting lineage-specific shifts in diversification: A proper likelihood approach” Syst Bio 2020? (I have to admit I haven’t read that paper in detail yet). Check also L305-306, 448-450, 620-621, and L304-305 and 933, 1095 of the Supp Info concerning the analytical solution of BAMM.

This statement still holds. The cited paper provides analytical likelihood equations for some special cases of a variant of the BAMM model. We now discuss this paper in the Supplementary material when we present the BAMM model.

L614-617: ‘The normalization constant estimates matched for LSBDS (Supplementary Fig. 8) and for ClaDS0 (Supplementary Fig. 7) but not for ClaDS1 and ClaDS2.’ This sentence

seems a little bit too negative given results from Fig7. A sentence that would better represent results from Fig. 7 would be ‘The normalization constant estimates matched for LSBDS (Supplementary Fig. 8) and for ClaDS0 (Supplementary Fig. 7); for ClaDS1 and ClaDS2, they matched for low values of lambda and mu (or epsilon), but not for larger values (Supplementary Fig. 7).’

We agree with the reviewer and have adopted the proposed wording.

*Section S3.2. L217-220 ‘Our simulations are weighted with the appropriate conversion factor to generate the density for labelled and unoriented trees. Thus, the normalization constants we compute are directly comparable to the likelihoods computed using the standard analytical equations established for the simple diversification models, such as CRBD’. This comparison of normalization constant to the analytical likelihood is used in section S7 to verify the PPL scripts, and is certainly correct given the results, but could the authors explain this a little bit more to the reader? As I understand it with have: likelihood of labelled reconstructed tree = normalization constant * likelihood of unlabeled oriented tree. So how do we end up having normalization constant = likelihood of unlabeled oriented tree? This will also help understanding why the normalization constant provides model evidence (SI L495).*

We think the problem here is that 'normalization constant' is a general term that can refer to different things. Here, we mean the normalization constant in Bayes' theorem, the marginal likelihood, that provides a measure of model evidence. We now make this explicit in the texts on SI L495 and SI L217-220.

SI L353: “Upon examination of the empirical results published in the same paper, we concluded that this choice is overly conservative.” Can the authors specify which empirical results led them to this conclusion? By overly conservative, do they mean that the choice of the prior in the Maliet et al. paper leads to an underestimation on rate heterogeneity?

Yes, we mean that the choice of the prior in Maliet et al. could lead to an underestimate of rate heterogeneity. This is now explained in the Supplementary material.

SI L1043. Did the authors mean “positively related” instead of “inversely related”? (simple diversification models appear more adequate in old trees).

We thank the reviewer for spotting this error. It is corrected in the revised version of the manuscript.

SI L1080: I guess the authors mean “higher marginal likelihoods”?

Yes, we do. This is clarified in the revised version of the manuscript.

Here is a list of parts of the paper I did not really understand, in case the authors could clarify:

- L223: ‘If such constructs are provided by PGM-based software, they are only executed when the model is initiated; they are not part of the model description itself.’

This sentence has been deleted during the revision. We agree that it was difficult to understand and potentially confusing.

- SI L784-786: ‘Since, for every execution of the program, there is a different number of hidden speciation events on each branch in the observed tree, this will cause the SMC particles to get out of sync at resampling points, so that we will be comparing particles that can be at very different points in the simulation. ‘

We added a sentence that should clarify what we mean.

- SI Listing 3 is the description of a complete WebPPL script that one needs to understand to have a good idea of how the inference works. However, I found it hard to follow. I was particularly lost with the computation of the weights in the simulation functions. As mentioned on Lines 588-589, the simulation function does not return anything, but it weights the sampled parameters by conditioning the simulation on the observed tree, however I don’t see how/where this weighting is done. Where does the computation of the weights $\ln Prob1$, $\ln Prob2$ and $\ln Prob3$ mentioned on Line 591-593 occur in the script?

We apologize for causing this confusion. The text in the original version of the manuscript partly described the aligned script and not the simpler naive script that was shown in the

listing, where there is no accumulation of likelihood factors (the `lnProbX` variables mentioned by the reviewer are not present in the naive script, only in the aligned script). We have now corrected the text so that it matches the naive script in the code listing.

REVIEWERS' COMMENTS:

Reviewer #1 (Remarks to the Author):

I appreciate the authors' extensive response to the issues I raised and also apologize for an oversight I have made in mischaracterizing the software tools that were used in their analyses. The majority of my comments centered around comparisons of the breadth and capabilities of probabilistic modeling frameworks often used and the authors have done a good job in clarifying the most important points. I am also very hopeful that the authors bring future work on cross-tree sampling to fruition soon, though I agree that it is perhaps beyond the scope of this manuscript to include it. In my view, the authors have satisfactorily addressed my comments and the work is ready for publication.

Reviewer #3 (Remarks to the Author):

The authors have taken all comments very seriously and done a great job at revising their manuscript. I have only minor comments, but I think two of them (the first 2) deserve a little more attention. I am looking forward to seeing the paper published.

1. Explanation for the observed low extinction rates (L562-567 & Supplement 10.6). "This could be due in part to systematic biases in the sampling of the leaves in the observed trees^{40,41}, a problem that could be addressed by extending our PPL model scripts. Such sampling biases may also partly explain the strong support for slowing diversification rates²³." Although I agree that unaccounted-for diversified sampling will lead to an underestimation of extinction rates (and signals of diversification slowdown), it does not seem to be a plausible explanation here (i.e. for the birds phylogeny). As acknowledged by the authors (L1282-1283 in the SM), the bird trees are fairly complete with no obvious sampling bias. The authors reasoning is that there might be "Unacknowledged diversified sampling around the species level" linked to the phenomenon of protracted speciation. However, it makes sense that a species-level diversification model is applied to a species-level phylogeny (including only morphologically and genetically distinct species, and not incipient species or subspecies); if there is no diversified sampling at the species level, then I don't see how the argument holds. I suggest that the authors treat diversified sampling in the same way as they treated rate heterogeneity: a hypothesis that has been advanced for explaining low extinction rates but that does not seem to apply here. I think that despite a lot of work on this subject we still do not have a good explanation of why we always estimate low extinction rates.

2. A. Interpretation of the alpha parameter from ClaDS L. 352-354: "When $\alpha < 1$ (resp. $\alpha > 1$), the speciation rate decreases (resp. increases) exponentially on average, thus corresponding to $z < 0$ (resp. $z > 0$) in the case of the TDBD, TDB and BAMM models." We agree that seems intuitive, however we have seen that $\alpha < 1$ does not imply that the speciation rates decreases exponentially on average; it would be true in the absence of rate heterogeneity, but rate heterogeneity creates a "lineage selection effect" (lineages that by chance have high speciation rates leave more descendants) that renders this statement inaccurate. We show this in a recent paper that implements ClaDS using data augmentation available here (see page 11 and 3rd paragraph of page 12). Note that this comment is distinct from the earlier comment in my previous review on the use of alpha vs m. B. Related to the later comment, given the above, I am not sure that the argument of using α because "it is readily comparable to the z parameter in the TDB(D) and BAMM models" holds. The authors also make the argument that computing alpha is more efficient, but why would they not compute m from alpha and σ^2 ? I am not asking the authors to present results on m rather than on alpha if this is not their choice, I am just trying to understand and avoid confusion in the literature.

3. L49: "unobserved side branches that have gone extinct" \diamond add "or have not been sampled".

4. L110-111: "despite the difficulties in developing good inference algorithms for them" \diamond replace by "some of them" (there are good inference algorithms for all of them except BMM).

5. Paragraph starting from L. 418: See the ref provided in point 2, there are now two different implementations of ClaDS. Maybe clarify for the reader: "The ClaDS models were initially implemented in the R package RPANDA 25 , using a combination of advanced numerical solvers and MCMC simulation¹⁵. A new implementation of ClaDS2 in Julia instead relies on data augmentation (Maliet & Morlon BioRxiv). L571-575: "Our verification experiments (Supplementary Section 7) suggest that the light-weight PPL implementations of ClaDS1 and ClaDS2 provide more accurate computation of likelihoods than the thousands of lines of code developed in the first implementation of these models (Maliet et al. NEE 2019).

Response to second review

We thank reviewer 3 for valuable additional comments on our revised manuscript. We provide detailed answers below.

Reviewer #3 (Remarks to the Author)

The authors have taken all comments very seriously and done a great job at revising their manuscript. I have only minor comments, but I think two of them (the first 2) deserve a little more attention. I am looking forward to seeing the paper published.

1. Explanation for the observed low extinction rates (L562-567 & Supplement 10.6). “This could be due in part to systematic biases in the sampling of the leaves in the observed trees^{40,41}, a problem that could be addressed by extending our PPL model scripts. Such sampling biases may also partly explain the strong support for slowing diversification rates²³.” Although I agree that unaccounted-for diversified sampling will lead to an underestimation of extinction rates (and signals of diversification slowdown), it does not seem to be a plausible explanation here (i.e. for the birds phylogeny). As acknowledged by the authors (L1282-1283 in the SM), the bird trees are fairly complete with no obvious sampling bias. The authors reasoning is that there might be “Unacknowledged diversified sampling around the species level” linked to the phenomenon of protracted speciation. However, it makes sense that a species-level diversification model is applied to a species-level phylogeny (including only morphologically and genetically distinct species, and not incipient species or subspecies); if there is no diversified sampling at the species level, then I don’t see how the argument holds. I suggest that the authors treat diversified sampling in the same way as they treated rate heterogeneity: a hypothesis that has been advanced for explaining low extinction rates but that does not seem to apply here. I think that despite a lot of work on this subject we still do not have a good explanation of why we always estimate low extinction rates.

Response: We agree with the reviewer that diversified sampling should be described as a possible contributing factor, among several, and that the reason for the underestimation of extinction rates is still not adequately understood. However, we still think there is diversified sampling bias in the bird tree, and we do think that missing incipient species could have an effect on the estimates of diversification rates.

We now have added simulation results to the Supplementary Material to show that there is indeed diversified sampling bias in the bird data. There are 9,993 species in the full bird tree, and only 6,670 of those were sequenced. If the latter were a random sample of the former, the number of genera covered by the sequenced set should not be significantly different from that

of a similarly sized random sample of species. However, if the sample is diversified, we might expect to see more genera in the sequenced set than expected by chance.

Simulations show that there are indeed significantly more genera in the sequenced set than expected by chance. The sequenced set includes 1,880 genera, while 10,000 random samples of 6,670 bird species included on average 1,759 genera (range 1,703 to 1,808). Thus, there is a distinct diversified sampling bias. The degree to which this sampling bias affects extinction rate estimates requires further analysis, which we think would be beyond the scope of the current paper.

The problem with incipient species or genetically separate populations, we think, is that some of them do represent true species, we just do not have enough information at the present to clearly identify which ones. Thus, under-sampling of incipient species is problematic also for birth-death models at the species level, and the effect is similar to that of diversified sampling.

In the revised version of the manuscript, we have expanded on these topics in the Supplementary material. We have also emphasized there and in the Discussion in the main text that there are other potential explanations for underestimated extinction rates and slowing diversification rates. Finally, throughout the discussion of these factors, we have ensured that we say that diversified sampling "could" or "may" contribute to low extinction rate estimates or support for slowing diversification rates.

2. A. Interpretation of the alpha parameter from ClaDS L. 352-354: "When $\alpha < 1$ (resp. $\alpha > 1$), the speciation rate decreases (resp. increases) exponentially on average, thus corresponding to $z < 0$ (resp. $z > 0$) in the case of the TDBD, TDB and BAMM models." We agree that seems intuitive, however we have seen that $\alpha < 1$ does not imply that the speciation rates decreases exponentially on average; it would be true in the absence of rate heterogeneity, but rate heterogeneity creates a "lineage selection effect" (lineages that by chance have high speciation rates leave more descendants) that renders this statement inaccurate. We show this in a recent paper that implements ClaDS using data augmentation available here (see page 11 and 3rd paragraph of page 12). Note that this comment is distinct from the earlier comment in my previous review on the use of alpha vs m. B. Related to the later comment, given the above, I am not sure that the argument of using α because "it is readily comparable to the z parameter in the TDB(D) and BAMM models" holds. The authors also make the argument that computing alpha is more efficient, but why would they not compute m from alpha and sigma2? I am not asking the authors to present results on m rather than on alpha if this is not their choice, I am just trying to understand and avoid confusion in the literature.

Response. This is an excellent comment, and we essentially agree. The dynamics of the ClaDS models is distinctly different from that of the TDBD model, and we should not have

suggested that the alpha parameter of ClaDS is readily comparable to the z parameter of TDBD. It seems to us now that the dynamics of the ClaDS models is quite complex, and that neither the interpretation of alpha nor of m is straightforward.

Assume, for instance, that you set both alpha and m to 1.0, and then increase sigma gradually from 0. Assume further that we focus on a single lineage and look at the expectation of lambda after some finite period of time, keeping mu constant (as in ClaDS0 or ClaDS1). Somewhat surprisingly, when m=1, the ending lambda will on average be slightly smaller than the starting lambda because of the "stopping phenomenon", that is, because it is more likely to reach the end of the time segment when lambda is low. The same phenomenon occurs when alpha=1, but this is outweighed by the fact that the expectation of m, when alpha is set to 1, is larger than 1, specifically it is $m = \alpha * \exp(\sigma^2/2)$, as pointed out previously by the reviewer. The net effect for alpha=1 is a slight but distinct increase in the expectation of lambda over the time segment. In both cases there is also a lineage selection effect, which tends to increase lambda across lineages as demonstrated by the reviewer in the recent bioRxiv paper.

In the revised version of the manuscript, we have extensively rewritten the main text and the supplementary text on the parameterization of the ClaDS models. We justify the choice of the alpha parameterization based on the fact that it allows us to specify a conjugate prior that makes SMC inference efficient. We also point out that the ClaDS models have complex dynamics that differs considerably from superficially similar models, such as TDBD, TDB and BAMM. We discuss these complex dynamics in the supplementary text, and we refer to the recent bioRxiv paper in this context both in the main text and in the supplementary text. Finally, we point out in the supplementary text that it is straightforward to estimate posterior distributions on m using our scripts, either by converting alpha to m in the scripts themselves before running the analysis, or by converting the sampled alpha values to m values after the analysis has completed.

3. L49: *"unobserved side branches that have gone extinct"* - add *"or have not been sampled"*.

Response: Excellent suggestion, this has been added.

4. L110-111: *"despite the difficulties in developing good inference algorithms for them"* - replace by *"some of them"* (there are good inference algorithms for all of them except BAMM).

Response: Modified as suggested.

5. Paragraph starting from L. 418: See the ref provided in point 2, there are now two different implementations of ClaDS. Maybe clarify for the reader: “The ClaDS models were initially implemented in the R package RPANDA 25 , using a combination of advanced numerical solvers and MCMC simulation¹⁵. A new implementation of ClaDS2 in Julia instead relies on data augmentation (Maliet & Morlon BioRxiv). L571-575: “Our verification experiments (Supplementary Section 7) suggest that the light-weight PPL implementations of ClaDS1 and ClaDS2 provide more accurate computation of likelihoods than the thousands of lines of code developed in the first implementation of these models (Maliet et al. NEE 2019).

Response: Text amended as suggested.